# FLUENT ALIGNMENT WITH DISFLUENT JUDGES: POST-TRAINING FOR LOWER-RESOURCE LANGUAGES

**David Samuel**    **Lilja Øvrelid**    **Erik Velldal**    **Andrey Kutuzov**
Language Technology Group, University of Oslo
davisamu@ifi.uio.no

## ABSTRACT

We propose a post-training method for lower-resource languages that preserves the fluency of language models even when aligned by disfluent reward models. Preference optimization is now a well-researched topic, but previous work has mostly addressed models for English and Chinese. Lower-resource languages lack both datasets written by native speakers and instruction-tuned language models capable of generating fluent synthetic data. To address this, we focus on developing a fluent preference-aligned language model without any instruction-tuning data in the target language. Our approach uses an on-policy training method, which we compare with two common alternatives: supervised finetuning on machine-translated data and multilingual finetuning. We conduct a case study on Norwegian Bokmål and evaluate fluency through native-speaker assessments. The results show that the on-policy aspect is crucial and outperforms the alternatives without relying on any hard-to-obtain data.

## 1 INTRODUCTION

Instruction-tuning and preference-optimization have become a cornerstone of modern language models, enabling base models to follow instructions and engage in helpful dialogue. However, this progress has been largely confined to high-resource languages like English and Chinese, which benefit from extensive human-written datasets and sophisticated language models capable of generating fluent synthetic data. Lower-resource languages face a fundamental challenge: they lack both instruction datasets written by native speakers and fluent instruction-tuned models that could generate high-quality training data (Guo et al., 2025). This work addresses a critical question for the democratization of language technology: how can we create fluent preference-aligned language models for lower-resource languages without any instruction-tuning dataset in the target language?[1]

Current approaches to post-training language models for lower-resource languages mostly rely on static and predefined instruction-tuning datasets (Suzuki et al., 2023; Chouikhi et al., 2024; Lim et al., 2025), which are usually machine-translated from English (Pipatanakul et al., 2023; Santilli & Rodolà, 2023; Ranaldi & Pucci, 2023; Üstün et al., 2024; Nguyen et al., 2024; Bari et al., 2025; Zosa et al., 2025). While this approach shows promising results when evaluated on standard NLP benchmarks, the translation process introduces subtle linguistic artifacts – *translationese* – that make the resulting models disfluent in the target language (Yu et al., 2022; Dutta Chowdhury et al., 2022). Translationese is produced even by professional human translators, and machine-translation models are negatively impacted by it to an even greater extent (Bizzoni et al., 2020). Concurrent work by Kunz (2026) has demonstrated that even short exposure to such data leads to a rapid loss of fluency. Thus, we believe that post-training for lower-resource languages needs to shift away from such inherently flawed data.

Recent advances in reinforcement learning from AI feedback (RLAIF; Bai et al., 2022) offer a potential solution to this challenge. In on-policy reinforcement learning, the model learns from its own generated responses rather than from fixed datasets. This means we can potentially avoid exposing it to disfluent text altogether. The key insight is that a model that has learned fluent generation through extensive pretraining on native texts can maintain this fluency as long as it is never trained on unnatural examples during the alignment phase.

---

[1]*Fluency* refers to the linguistic quality of text that makes it natural, grammatical, and easy to read. It should read as though written by a native speaker. It is independent of other qualities such as factual accuracy.

In this work, we propose a fluency-aware post-training method that leverages on-policy reinforcement learning to align language models for lower-resource languages without compromising their linguistic naturalness. Crucially, we never train the model on any translated responses, preserving the fluent generation capabilities learned during pretraining. We demonstrate that even a disfluent judge model can successfully guide a fluent policy, as long as the judge understands the target language well enough to evaluate response quality.

To validate this approach, we need a language that is genuinely under-resourced for post-training yet has enough infrastructure for rigorous evaluation. *Norwegian (Bokmål)*, a language with about 5 million speakers, fits this profile well: it lacks manually written instruction datasets, yet the research community has pretrained several Norwegian language models that can be leveraged for post-training. Lastly, we employed five native Norwegian speakers for accurate manual evaluation of fluency.

In the native-speaker evaluations, our on-policy approach is preferred over supervised finetuning on machine-translated data in 67.5% of pairwise comparisons while requiring no instruction-tuning data in Norwegian. Our main contributions are:

- We propose an on-policy reinforcement learning method for post-training in lower-resource languages that maintains fluency without requiring any instruction datasets in the target language.
- We demonstrate through extensive human evaluation with native Norwegian speakers that on-policy training produces more fluent models than supervised finetuning on translated data.
- We show that fluent aligned models can be bootstrapped using disfluent judges, enabling post-training for languages without existing fluent instruction-tuned models.
- We provide comprehensive ablations revealing the critical importance of avoiding any exposure to translated responses – even in small quantities – during training.

## 2 FLUENCY-AWARE POST-TRAINING

This section describes our proposed method for post-training language models on a target lower-resource language without any dedicated dataset in that language. The overall approach consists of three stages: i) pretraining on the target language, ii) short SFT alignment on English, and iii) on-policy alignment on the target language.

The key principle is to *never train the language model on any unnatural text*.

**Pretraining on target language**  The first stage is essential to ensure that the base language model learns all necessary linguistic knowledge and is able to generate fluent outputs in the target language. Our study focuses on the subsequent training stages and does not cover this pretraining stage – that has already been studied in detail by Gururangan et al. (2020); Ibrahim et al. (2024); Kim et al. (2024); Samuel et al. (2025). Specifically for Norwegian, we build upon the multi-stage continual pretraining from Samuel et al. (2025).

**Short SFT alignment on English**  The second stage is crucial for teaching the model to respond to user prompts and follow the specific chat format (Appendix E.1). Typically, supervised finetuning would be done on a carefully curated set of conversations in the target language, but such resources are usually not available for lower-resource languages. We therefore opt for a short alignment on a small high-quality English dataset. Specifically, we use the 1 000 curated prompt-response pairs from LIMA (Zhou et al., 2023) and train the base model on this dataset for one epoch. The short training ensures that the model does not catastrophically forget its innate knowledge of the target language.

**On-policy alignment on target language**  Finally, in the third stage, the language model is aligned to respond in a helpful, truthful and safe way – without losing the fluency it acquired during the initial pretraining stage. This is achieved by training with online on-policy reinforcement learning where the model is trained solely on responses sampled from itself. In this way, the model is never pushed away from the subspace of fluent outputs it has learned to prefer during pretraining. A key observation is that we do not need to train any reward model as it suffices to use an LLM-as-a-judge system to provide the reward signal (Zheng et al., 2023); *as long as the judge understands the target language, it does not have to be fluent to produce a fluent policy.* We evaluate this hypothesis later in Section 4.2. This effectively allows us to bootstrap fluent models in languages without any instruction datasets and without any existing fluent language models.

## 2.1 Online on-policy reinforcement learning

We now describe the components of our on-policy training: the reward signal, the policy gradient objective, KL regularization, and the distributed implementation. It is important to note that any implementation of online on-policy training with feedback from an LLM-as-a-judge system should work comparably well in terms of fluency of the final policy network – our implementation is chosen to be simple and comparable to the baseline approach of supervised finetuning, as detailed later in Section 3. The algorithm does not directly optimize fluency: as detailed below, the reward signal captures the overall quality of responses based on the standard criteria such as correctness, safety or conciseness – without explicitly targeting fluency (Appendix E).

**Reward model**  The standard approach to on-policy alignment – reinforcement learning from human feedback (RLHF; Christiano et al., 2017; Stiennon et al., 2020; Ouyang et al., 2022) – first trains a Bradley-Terry reward model on a preference dataset from the target domain. Since we are restricted to a lower-resource language, we assume that such a dataset is not available – instead, we rely on direct reinforcement learning from AI feedback (d-RLAIF; Lee et al., 2024).

In this scenario, we only use the domain knowledge to create a prompt template (constitution) that clearly guides a multilingual language model to judge the quality of responses. As evident from later results in Section 4.2, this setup provides enough signal that even a disfluent judge can train a fluent and capable policy – as long as the judge has some level of understanding of the target language.

**Objective function**  Let us start with defining the objective $\mathcal{J}$ to maximize during this post-training stage. The objective in Equation (1) states that we want to find parameters $\boldsymbol{\theta}$ of our language model $\pi_{\boldsymbol{\theta}}$ (the policy model) that maximize the reward $r$ given to a prompt $\boldsymbol{x} = (x_1, x_2, \ldots x_{|\boldsymbol{x}|})$ and its corresponding response $\boldsymbol{y} = (y_1, y_2, \ldots y_{|\boldsymbol{y}|})$ sampled from the policy $\pi_{\boldsymbol{\theta}}$:

$$\underset{\boldsymbol{\theta}}{\arg\max}\, \mathcal{J}(\boldsymbol{\theta}) \stackrel{\text{def}}{=} \underset{\boldsymbol{\theta}}{\arg\max}\, \underset{\boldsymbol{x} \sim \mathcal{D},\, \boldsymbol{y} \sim \pi_{\boldsymbol{\theta}}(\cdot|\boldsymbol{x})}{\mathbb{E}}\, r(\boldsymbol{x}, \boldsymbol{y}). \tag{1}$$

Following the majority of works on LM alignment, we optimize the objective with *policy gradient* methods (Williams, 1992; Sutton et al., 2000) that perform gradient descent on $-\nabla_{\boldsymbol{\theta}}\mathcal{J}(\boldsymbol{\theta})$ using online on-policy samples $\boldsymbol{y}$ from the policy model $\pi_{\boldsymbol{\theta}}$:

$$-\nabla_{\boldsymbol{\theta}}\mathcal{J}(\boldsymbol{\theta}) = -\underset{\boldsymbol{x} \sim \mathcal{D},\, \boldsymbol{y} \sim \pi_{\boldsymbol{\theta}}(\cdot|\boldsymbol{x})}{\mathbb{E}}\, r(\boldsymbol{x}, \boldsymbol{y})\, \nabla_{\boldsymbol{\theta}} \log \pi_{\boldsymbol{\theta}}(\boldsymbol{y}|\boldsymbol{x}). \tag{2}$$

Directly using Equation (2) for training gives us the REINFORCE algorithm (Williams, 1992). To increase its convergence speed and stability, we modify the rewards and optimize advantages $A(\boldsymbol{x}, \boldsymbol{y})$ instead of $r(\boldsymbol{x}, \boldsymbol{y})$. Following REINFORCE WITH BASELINE (Weaver & Tao, 2001), we subtract the baseline score $b(\boldsymbol{x})$, and following Karpathy (2016), we further normalize by the dispersion factor $s(\boldsymbol{x})$; giving us $A(\boldsymbol{x}, \boldsymbol{y}) \stackrel{\text{def}}{=} (r(\boldsymbol{x},\boldsymbol{y})-b(\boldsymbol{x}))/s(\boldsymbol{x})$. While these additional factors are often estimated by separately trained models (Konda & Tsitsiklis, 2000; Schulman et al., 2017; Christiano et al., 2017), we choose a more straightforward approach and estimate them as the sample mean and the sample standard deviation over $G$ responses (Kool et al., 2019; Ahmadian et al., 2024; Shao et al., 2024):

$$\hat{A}(\boldsymbol{x}, \boldsymbol{y}) \stackrel{\text{def}}{=} \frac{r(\boldsymbol{x}, \boldsymbol{y}) - \text{mean}\big(\{r(\boldsymbol{x}, \boldsymbol{y}^{(i)})\}_{i=1}^{G}\big)}{\text{std}\big(\{r(\boldsymbol{x}, \boldsymbol{y}^{(i)})\}_{i=1}^{G}\big)}. \tag{3}$$

Putting Equation (2) and Equation (3) together, we can define the loss function $\mathcal{L}(\boldsymbol{\theta}, \boldsymbol{x})$ of a single query $\boldsymbol{x}$ for optimizing the parameters $\boldsymbol{\theta}$ of a policy $\pi_{\boldsymbol{\theta}}$. One important detail is to account for the potential length bias (Liu et al., 2025) when computing the log-likelihood $\log \pi_{\boldsymbol{\theta}}(\boldsymbol{y}|\boldsymbol{x})$ as $\sum_{j=1}^{|\boldsymbol{y}|} \log \pi_{\boldsymbol{\theta}}(y_j|\boldsymbol{x}, \boldsymbol{y}_{<j})$ and normalizing by $1/|\boldsymbol{y}|$. As proposed by Rastogi et al. (2025), the length bias can be mitigated by dividing by the total length of responses $\{\boldsymbol{y}^{(i)}\}_{i=1}^{G}$ to a prompt $\boldsymbol{x}$. Then the token-level loss function becomes:

$$\nabla_{\boldsymbol{\theta}}\mathcal{L}(\boldsymbol{\theta}) \stackrel{\text{def}}{=} -\underset{\boldsymbol{x} \sim \mathcal{D},\, \{\boldsymbol{y}^{(i)}\}_{i=1}^{G} \sim \pi_{\boldsymbol{\theta}}(\cdot|\boldsymbol{x})}{\mathbb{E}} \left[ \frac{1}{\sum_i |\boldsymbol{y}^{(i)}|} \sum_{i=1}^{G} \hat{A}(\boldsymbol{x}, \boldsymbol{y}^{(i)}) \sum_{j=1}^{|\boldsymbol{y}^{(i)}|} \nabla_{\boldsymbol{\theta}} \log \pi_{\boldsymbol{\theta}}(y_j^{(i)}|\boldsymbol{x}, \boldsymbol{y}_{<j}^{(i)}) \right], \tag{4}$$

Note that unlike most current work on LM alignment, we do not modify the loss function to account for sampling from a proximal policy (by clipping outlier samples and importance sampling) as in PPO

(Schulman et al., 2017). As detailed below, the synchronous parallelization makes all samples almost fully on-policy and we have found no benefit in moving away from the simple REINFORCE-like loss function in Equation (4).

**KL-divergence regularization**   A well known issue of policy gradient methods in RLHF is that they optimize model-based rewards instead of the true (and unknown) rewards. When the policy is modeled by a large language model with billions of parameters, it can very quickly find shortcuts in the reward model and *reward-hack* its measured performance while degrading its true performance. The most common way to mitigate this issue is to add a soft constraint to the optimization problem that pushes the policy to stay close to its original state – the distance is usually measured by the KL divergence and the strength of the constraint is parametrized by $\beta$ (Kullback & Leibler, 1951; Jaques et al., 2019):

$$\underset{\boldsymbol{\theta}}{\arg\max}\, \mathcal{J}(\boldsymbol{\theta}) \overset{\text{def}}{=} \underset{\boldsymbol{\theta}}{\arg\max}\, \underset{\boldsymbol{x} \sim \mathcal{D},\, \boldsymbol{y} \sim \pi_{\boldsymbol{\theta}}(\cdot|\boldsymbol{x})}{\mathbb{E}} \Big[ r(\boldsymbol{x}, \boldsymbol{y}) - \beta D_{\text{KL}}[\pi_{\boldsymbol{\theta}}(\cdot \mid \boldsymbol{x}) \,\|\, \pi_{\boldsymbol{\theta}_{\text{ref}}}(\cdot \mid \boldsymbol{x})] \Big]. \quad (5)$$

In order to optimize the objective $\mathcal{J}$ according to the new definition instead of the simpler one from Equation (1), we need to introduce an additional loss term $\mathcal{L}_{KL}$ (weighted by $\beta$) that will push the trained policy $\pi_{\boldsymbol{\theta}}$ closer to the output distribution $\pi_{\boldsymbol{\theta}_{\text{ref}}}$.

The problem with KL divergence is that its exact computation is intractable in most cases. This means that we have to approximate it in practice, the most common way is to simply use the already sampled prompts with responses and compute a direct Monte-Carlo estimate of $\mathbb{E}_{\boldsymbol{x} \sim \mathcal{D},\, \boldsymbol{y} \sim \pi_{\boldsymbol{\theta}}(\cdot|\boldsymbol{x})}\big[\log \pi_{\boldsymbol{\theta}}(\boldsymbol{y}|\boldsymbol{x})/\pi_{\boldsymbol{\theta}_{\text{ref}}}(\boldsymbol{y}|\boldsymbol{x})\big]$, as done in the seminal RLHF work by Stiennon et al. (2020). While straightforward, this approximation is very rough and ill-behaved, even becoming negative sometimes.

These estimates use only a small fraction of information available in the probability distributions given by $\pi_{\boldsymbol{\theta}}(\cdot|\boldsymbol{x}, \boldsymbol{y}_{<i})$ – only the single value of $\pi_{\boldsymbol{\theta}}(y_i|\boldsymbol{x}, \boldsymbol{y}_{<i})$. Instead, we can get provably tighter estimates when we Rao-Blackwellize the Monte-Carlo estimation by using the full next-token distributions over the vocabulary $\mathcal{V}$. Amini et al. (2025) prove that this estimation is unbiased and has lower variance than the standard Monte-Carlo estimation.

$$\mathcal{L}_{KL}(\boldsymbol{\theta}) \overset{\text{def}}{=} \underset{\boldsymbol{x} \sim \mathcal{D},\, \boldsymbol{y} \sim \pi_{\boldsymbol{\theta}}(\cdot|\boldsymbol{x})}{\mathbb{E}} \Bigg[ \sum_{i=1}^{|\boldsymbol{y}|} \underbrace{\sum_{w=1}^{|\mathcal{V}|} \pi_{\boldsymbol{\theta}}(y_i = w \mid \boldsymbol{x}, \boldsymbol{y}_{<i}) \cdot \log \frac{\pi_{\boldsymbol{\theta}}(y_i = w \mid \boldsymbol{x}, \boldsymbol{y}_{<i})}{\pi_{\boldsymbol{\theta}_{\text{ref}}}(y_i = w \mid \boldsymbol{x}, \boldsymbol{y}_{<i})}}_{D_{\text{KL}}[\pi_{\boldsymbol{\theta}}(\cdot \mid \boldsymbol{x}, \boldsymbol{y}_{<i}) \,\|\, \pi_{\boldsymbol{\theta}_{\text{ref}}}(\cdot \mid \boldsymbol{x}, \boldsymbol{y}_{<i})]} \Bigg]. \quad (6)$$

The computational overhead of the Rao-Blackwellized estimate is negligible because it still requires only a single forward-backward pass through the policy model that is done even without any KL regularization. Another benefit of regularizing the full output distribution is that it eliminates the need for another loss term for regulating the output entropy – as used in most RLHF works – further simplifying the training method.

**Distributed setup**   As opposed to supervised finetuning, RL approaches need several language models to be fully materialized and used at the same time (the trained policy, the reference policy, the sampled policy and the reward model). In principle, these models should be run sequentially in a cycle (Figure 1), but that is inefficient in practice and the cycle needs to be broken and parallelized. As illustrated below in Figure 1, this can be achieved by postponing the update of the sampled policy – effectively turning the training slightly off-policy.

Unlike other approaches to distributed RL (Espeholt et al., 2018; Noukhovitch et al., 2025; Rastogi et al., 2025), our parallelization is completely synchronous. This can make the resources allocated for sampling underutilized (all workers have to wait until the longest response completes), but that does not impact the overall efficiency much as most resources are allocated to the reward models. On the other hand, our samples are unbiased (asynchronous approaches typically up-sample problems with short responses), and the completely synchronous training cycle simplifies the implementation, as well as the objective function. Since the samples are guaranteed to be off-policy by just three steps, we can still rely on on-policy training techniques without having to resort to more complicated and less stable proximal-policy methods such as PPO (Schulman et al., 2017).

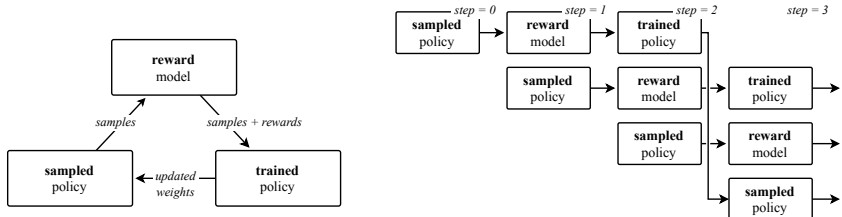

Figure 1: **Left: Reinforcement-learning cycle.** This diagram demonstrates the sequential nature of online RL training: each training step starts by sampling new responses from the policy model, followed by sampling response-judgments from the reward model, and then updating the weights of the policy model based on the sampled responses and rewards. **Right: Parallelization.** Breaking the cycle and postponing the update of the sampled policy allows for running all three models at the same time (vertically-aligned blocks are ran concurrently on different GPU nodes).

## 3  EXPERIMENT: ONLINE ON-POLICY TRAINING MAINTAINS FLUENCY

The main experiment of this paper aims to answer the central question: *Does online on-policy training produce more fluent language models than supervised finetuning on translated data?* To answer this, we designed the experiment to make on-policy training as similar to supervised finetuning as possible – using the same base model, the same training data, and the same number of training samples. As a case study for models trained on lower-resource language, we trained all models on Norwegian Bokmål; then we asked five native speakers to do pair-wise fluency comparisons of outputs generated from these models.

In total, we compare three post-training approaches represented by three language models that are based on the same pretrained model, as illustrated in Figure 2:

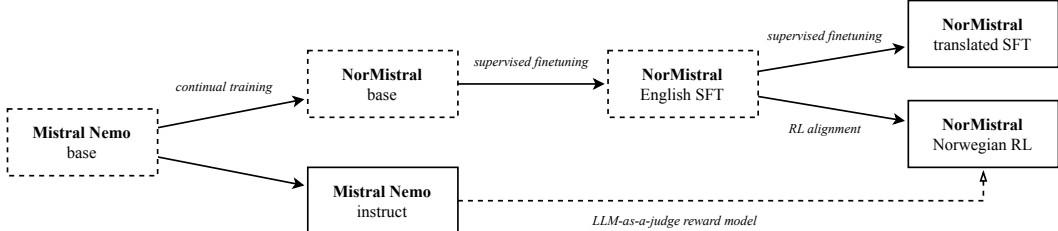

Figure 2: **Genealogy of the compared models.** The three models compared in the main fluency experiment (highlighted in bold boxes) all originate from a single base model – Mistral Nemo 12B (left). The official instruction-tuned version of this model also serves as the judge for our reinforcement-learning approach.

***Approach 1*: Norwegian RL**    We follow the method described in Section 2.1 when training this model. More specifically, we start from NorMistral 11B, a continually pretrained Norwegian base model from Samuel et al. (2025), and supervise-finetune it for a single epoch on the 1 000 English conversations from LIMA (Zhou et al., 2023). Then, in the final reinforcement-learning stage, we use the same dataset as the second approach: No Robots by Rajani et al. (2023). In each training step, we take 128 translated prompts from this dataset, sample a group of 8 responses for each prompt. Each response is then graded by a judge that also has access to the gold English response provided in the No Robots dataset – the judge prompt itself is attached in Appendix E. In order to rule out the possibility that the policy acquires its fluency from the judge model, we use Mistral Nemo 12B as the judge – a model with the same origin as the policy itself that is also evaluated in this experiment as the approach number three. More technical training details are described in Appendix C. Many of these choices are ablated later in Section 4 – for example, it is clear that Mistral Nemo is a poor reward model compared to larger language models, which should not, however, influence the fluency of the trained policy (Section 4.2).

*Approach 2*: **Translated SFT**    As illustrated in Figure 2, the translated-SFT model is initialized from the same checkpoint as the first approach. Then we also finetune it on the same machine-translated No Robots dataset – however in this scenario, we directly finetune the model on the translated responses by minimizing their conditional negative log-likelihood (in the first approach, the non-translated responses are only used as hints for the reward model). To translate the full No Robots dataset to Norwegian Bokmål, we use the Unbabel/Tower-Plus-72B model – currently the state-of-the-art open-weights translation model with explicit support for Norwegian (Rei et al., 2025). Other strong translation models are ablated later in Section 4.5.

*Approach 3*: **Mistral Nemo**    The last tested model is Mistral-Nemo-Instruct-2407 by Jiang et al. (2024a). Including it in this experiment serves three purposes: firstly, to give a reference baseline from an externally trained model; secondly, to test the fluency of the reward model that has been used in the first approach; and thirdly, to evaluate the performance of the standard multilingual – but English-focused – post-training that has been used in most major releases of the latest language models. This model shares the same origin as the previous two. However, while its weights are openly available, no details about its post-training process have been published.

**Manual annotation of generated outputs**    Fluency is a language feature that is difficult to accurately measure using statistical models, but it should be relatively straightforward to judge for native speakers of that language. We therefore hired five research assistants, all native Norwegian speakers, to compare the fluency of responses generated for a pool of prompts. The prompts are gathered from the Norwegian Bokmål mimir-instruct dataset from de la Rosa et al. (2025), which consists of artificially generated responses to prompts written by native Norwegian speakers. Since we use this data primarily as seeds for diverse generated outputs to evaluate for fluency, we select the 100 queries with the longest gold responses as the seed prompts; to ensure sufficiently long responses for meaningful fluency comparison. Then we sample a single response from each of the three evaluated models – using Nucleus sampling with temperature of 0.5, top_k of 64 and top_p of 0.9 (Holtzman et al., 2020). The responses are formed into 3 pairs per prompt (one for each model pair), yielding 300 pairs in total. These pairs are then rated by the annotators in the A/B testing fashion – *Which is more fluent: response A, response B, or are they equally fluent?* Each annotator is presented with the 300 pairs of responses in a randomized order; going through the full set of pairs took each annotator roughly 15–20 hours. We give more details about the actual annotation guidelines and the overall process in Appendix D. The resulting dataset (with anonymized annotator names) is published online at https://hf.co/datasets/ltg/normistral-fluency-annotation.

**Results**    We show the resulting win-rates from the manual annotation in Table 1. These win-rates are calculated by going through all pair-wise comparisons and either giving the winning model (preferred by more annotators) a single point or giving both models half a point in case of a tie – so-called $1/0.5/0$ method of aggregation (Copeland, 1951).

Table 1: **Model win-rates according to the manual fluency annotation.** The values show the win-rate percentages of the row-wise models over the column-wise models. The last column aggregates the win-rates of each model.

| Model | on-policy RL | translated SFT | Mistral Nemo | average |
|---|---|---|---|---|
| on-policy RL | — | 67.5 | 91.8 | 79.7 |
| translated SFT | 32.5 | — | 87.5 | 60.0 |
| Mistral Nemo | 8.2 | 12.5 | — | 10.3 |

The results clearly show that the most-preferred responses are from the on-policy training method, followed by translated SFT and then by Mistral Nemo. This supports our main claim since it demonstrates that the policy can indeed outperform its judge on fluency. It is worth noting that the first two approaches produced very fluent outputs and the fluency of $1/3$ of their pair-wise comparisons was agreed to be 'equal' – based on qualitative inspection, the difference between them mostly stems from infrequent traces of translationese in the SFT outputs (Appendix F).

## 4 FURTHER EVALUATIONS AND ABLATIONS

This section studies the effect of various post-training choices on the final performance in more detail. We scale up the manual fluency scoring from Section 3 by introducing an automatic fluency estimate, and also focus on more general performance of the trained models by incorporating Norwegian understanding and generation benchmarks.

### 4.1 EVALUATION METRICS

**Automatic fluency evaluation**  We have to rely on model-based measurement of fluency to lower the cost of evaluating all experiments in this section. The Norwegian fluency model is trained like a standard Bradley-Terry reward model (Stiennon et al., 2020) on a dataset $\mathcal{D}$ of paired preferred and non-preferred texts $(\boldsymbol{x}_w, \boldsymbol{x}_l) \in \mathcal{D}$. Specifically, we add a scalar linear head to a pretrained language model (`NorMistral 11B` in our case) and finetune it by minimizing the following loss:

$$\mathcal{L}_{fluency}(\boldsymbol{\theta}) \stackrel{\text{def}}{=} - \mathop{\mathbb{E}}_{\boldsymbol{x}_w, \boldsymbol{x}_l \sim \mathcal{D}} \Big[ \log \sigma \big( r_{\boldsymbol{\theta}}(\boldsymbol{x}_w) - r_{\boldsymbol{\theta}}(\boldsymbol{x}_l) \big) \Big]. \tag{7}$$

We create the training data by combining existing resources and newly synthesized texts. Firstly, we use the Norwegian ASK-GEC corpus of corrected language-learner essays (Jentoft, 2023) – from this corpus, we take all sentences with mistakes as the non-preferred texts and their (partially) corrected versions as the preferred texts. Secondly, we perform backtranslation with the OPUS collection of Norwegian-English machine translation models (Tiedemann et al., 2023): we sample a clean Norwegian sentence from the Norwegian Dependency Treebank (NDT; Solberg et al., 2014) or the Norwegian Review Corpus (NoReC; Velldal et al., 2018), then we sample one model to translate the sentence to English and then sample another model to translate it back to Norwegian; finally, the original sentence is cast as preferred and the backtranslated version as non-preferred. The ASK-GEC corpus trains the fluency scorer to take grammaticality into account while the second synthetic source focuses on translationese and lexical issues.

The fluency scorer can be directly evaluated by utilizing the manual annotations from the previous Section 3. Looking at all instances where the annotators agreed that one response is preferred over another response, the fluency scores agree with this ranking in 85.5% of cases. This agreement is even slightly higher than the agreement among annotators – when limiting their annotation to non-ties (for comparability), they agree with the consensus in 83.2% of cases, which highlights how subjective the notion of fluency can be. To lower the variance of the fluency score, we sample 16 responses from each evaluated model and average the scores. The raw scores are sigmoid-normalized into percentage values for clarity. When applied to the three approaches from the previous section, the fluency scores are 2.47 (92.2%) for *on-policy RL*, 1.94 (85.7%) for *translated SFT*, and 0.76 (65.3%) for *Mistral Nemo*, which correspond to the manual evaluation of these three models.

**Natural language understanding (NLU) evaluation**  We use the native-Norwegian reading-comprehension task NorQuAD to assess the level of Norwegian language understanding (Ivanova et al., 2023). In order to account for the variable and conversational outputs of instruction-tuned models, we evaluate the correctness of each generated response with an extra call to a judge, `Llama 3.3 70B` (Grattafiori et al., 2024), that compares it against the gold answer. This ensures that the evaluation is invariant to formatting variation of the generated outputs. We then report the average accuracy as the approximate Norwegian NLU score.

**Natural language generation (NLG) evaluation**  In order to assess the Norwegian generative abilities, we use two Norwegian benchmarks that were designed specifically for this purpose: NorRewrite and NorSummarize (Mikhailov et al., 2025); these benchmarks test the instruction-following, creative writing, and summarization abilities of Norwegian language models. We follow the original implementation of these benchmarks, which evaluates the quality of each response by comparing it pairwise with another response and automatically judging it with `Llama 3.3 70B`. We report the win-rate percentages of individual models against the smallest evaluated model, `Llama 3.1 8B` (Grattafiori et al., 2024).

## 4.2 FLUENT POLICY DOES NOT NEED A FLUENT REWARD MODEL

The main experiment has shown that even a disfluent judge (Mistral Nemo 12B) can produce a policy that is substantially more fluent than the judge itself. In this section, we investigate this phenomenon more thoroughly by checking that it is not an anomaly and that it holds for a diverse range of judges. First, we evaluate each judge on the three benchmarks described above: natural language understanding (NLU), generation (NLG), and fluency. Then, using a judge to provide the reward signal, we train a policy with the same method as in Section 3, and evaluate the policy on fluency – to assess the effect of the choice of judge.

Table 2: **The effect of the judge's knowledge of Norwegian on the trained policy.** The table shows the average Norwegian understanding, generation and fluency scores (Section 4) for different judges, and the fluency of the policy models trained with reward signals from these judges. The fluency scores are color-coded so that disfluent models are red and fluent models are blue.

| Judge | Judge performance | | | Fluency of trained policy |
|---|---|---|---|---|
| | NLU | NLG | Fluency | |
| Mistral Nemo 12B | 87.5 | 29.7 | 67.0 | 92.2 |
| Mistral Large 123B | 90.0 | 70.4 | 83.4 | **94.2** |
| Mixtral 8x22B | 91.3 | 20.2 | 70.9 | 92.1 |
| Llama 3.1 8B | 86.4 | 50.0 | 62.8 | 92.9 |
| Llama 3.3 70B | 90.7 | 57.7 | **84.2** | 93.5 |
| Qwen 2.5 14B | 89.6 | 43.5 | 39.0 | 93.1 |
| Qwen 2.5 32B | 91.7 | 59.9 | 43.2 | 93.9 |
| Qwen 2.5 72B | **92.0** | **75.2** | 50.7 | 92.9 |

**Results** We have evaluated language models of different sizes and different levels of Norwegian knowledge – three Mistral models (Jiang et al., 2024a;b), three Qwen models (Qwen et al., 2025), and two Llama models (Grattafiori et al., 2024) – the results of the evaluation are shown in Table 2. Comparing the fluency scores of judges with policies, there is no apparent correspondence (the Pearson correlation coefficient is 0.067); *the policies are fluent regardless of the (dis)fluency of their judge.* There is also no clear relation between the other two measures of judge quality on the resulting fluency. We hypothesize that fluency remains stable because the policy is trained exclusively on its own samples, which are fluent thanks to targeted pretraining. The choice of judge model appears to affect other response qualities but not fluency.

## 4.3 THE EFFECT OF TRAINING LENGTH ON FLUENCY

We further validate the claim that fluency is consistently stable for on-policy training by looking at the change in fluency score throughout training.

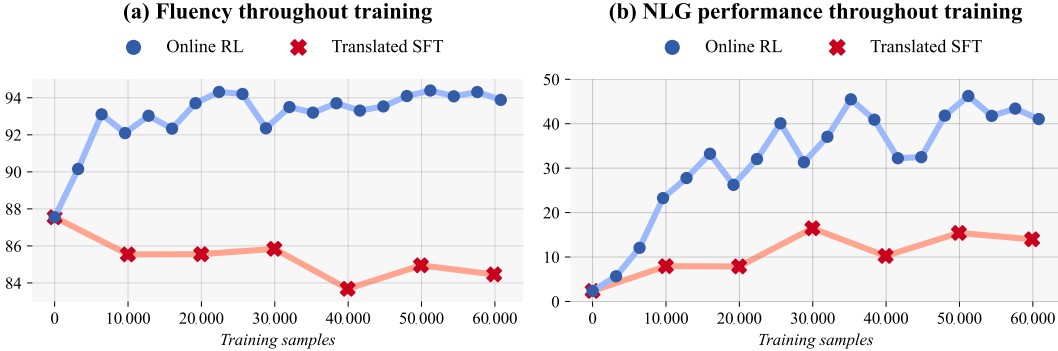

Figure 3: **Fluency and NLG scores throughout training.** We measure the performance score every 25 training steps for the reinforcement learning (in blue) and every epoch for the SFT training.

**Results**  Figure 3 shows the result of training a policy supervised by Mistral Large and evaluating its fluency and NLG performance every 25 training steps. *The fluency score remains stable around 93% after the initial convergence in the first 50 training steps.* Upon closer inspection, the lower initial fluency score of 87.5% can be explained by the increased likelihood of responding in English (not by disfluent Norwegian per se) triggered by the previous English SFT stage; then the model learns to consistently respond in Norwegian, which leads to the apparent increase in fluency (see Appendix F). On the other hand, when looking at the change of fluency throughout SFT training on translated Norwegian, we can see a clear decrease in fluency from the initial starting point, which can only be attributed to responding with consistent – but slightly disfluent – Norwegian. The longer reinforcement-learning run also allows us to look at the development of the directly optimized training metrics; these are appended in Appendix B.

### 4.4  THE IMPORTANCE OF NOT TRAINING THE POLICY ON ANY TRANSLATED RESPONSES

The previous experiment has highlighted the importance of the initial SFT stage and the ability of the policy model to recover from the sudden shift to English. As described earlier, we only train on the 1 000 examples (31 training steps) from the LIMA dataset (Zhou et al., 2023) in the initial SFT stage. The results in Table 3 suggest that training on more samples can slightly degrade fluency. Even clearer fluency deterioration is seen after machine-translating the dataset from English – *even a small amount of translated data can introduce a measurable amount of disfluencies in the final policy.*

### 4.5  IMPACT OF TRANSLATION QUALITY ON SFT PERFORMANCE

Even though we have selected Tower-Plus as the state-of-the-art model for translation from English to Norwegian (Rei et al., 2025), a possible explanation of our results is that the observed disfluencies are specific to this particular translation model rather than a general phenomenon. Thus, we translate the No Robots dataset with several alternative models, finetune on each translation, and compare the resulting fluency. We only test translation models that explicitly support English and Norwegian, which includes LLM-based translation models, massively multilingual models, and a traditional small model trained specifically for translation. The fluency scores in Table 4 confirm that Tower-Plus, as the largest available translation model, results in the most fluent policy.

Table 3: **The effect of the initial SFT stage.** We ablate various settings and measure the final fluency when starting reinforcement learning from such SFT checkpoints.

| SFT settings | RL fluency |
|---|---|
| English dataset (1 epoch) | **94.2** |
| English dataset (2 epochs) | 93.2 |
| English dataset (4 epochs) | 92.8 |
| Translated dataset (1 epoch) | 91.0 |

Table 4: **The effect of using different machine-translation models.** We train models on the translated dataset and measure their fluency.

| Translation model | Size | Fluency |
|---|---|---|
| Tower-Plus (Rei et al., 2025) | 72.7B | **85.7** |
| MADLAD-400 (Kudugunta et al., 2023) | 10.7B | 82.4 |
| Seed-X (Cheng et al., 2025) | 7.5B | 73.4 |
| NLLB-200 (Team et al., 2022) | 3.3B | 75.5 |
| OPUS Eng-Gem (Tiedemann et al., 2023) | 0.1B | 68.2 |

## 5  RELATED WORK

**Post-training on lower-resource languages**  As mentioned above, there have been numerous works that focused on supervised finetuning of language models for lower-resource languages (Suzuki et al., 2023; Pipatanakul et al., 2023, *inter alia*), substantially less attention has been paid to reinforcement learning for preference optimization in such languages (Lai et al., 2023; Dang et al., 2024). While fluency in less-spoken languages is a significant limitation of current language models, there has not been much interest from the research community in this topic – likely because accurate fluency evaluation is difficult. Dang et al. (2024) consider fluency as an important aspect of multilingual performance, but only include it as part of a general LLM-as-a-judge evaluation prompt, whose accuracy is not validated. Zhang et al. (2025) use human annotators to evaluate fluency of their models, but only after first back-translating the generated responses to English. A recent paper by Sainz et al. (2025) focuses on the effect of post-training configuration on fluency, similarly to

our work – they have fluency assessed by native (Basque) speakers, but only consider the effect of different data mixtures for supervised finetuning.

**Translationese and translation artifacts**    The phenomenon of translationese (Gellerstam, 1986; Baker et al., 1993; Toury, 1995) is well-documented in both human and machine translation (Koppel & Ordan, 2011; Volansky et al., 2013; Vanmassenhove et al., 2019; Bizzoni et al., 2020). Wang et al. (2023) have demonstrated that translationese in training data impacts performance in cross-lingual summarization, with models trained on translated data producing outputs that differ systematically from those trained on original text. This finding motivates our approach: rather than trying to mitigate translationese artifacts during training, we avoid them entirely by never training on any translated text. The concurrent work by Kunz (2026) studies the effect of fine-tuning on machine-translated Swedish and, similarly to us, found that it leads to a rapid loss of fluency.

**Weak-to-strong generalization**    The weak-to-strong generalization framework demonstrates that strong models consistently outperform their weak supervisors when fine-tuned on the labels generated by the weak supervisors (Burns et al., 2024). Following these empirical findings, Charikar et al. (2024) has proposed a theoretical frameworks that explains this phenomenon. It is particularly relevant to our setting: just as strong models can exceed the capabilities of weak supervisors, our Norwegian-pretrained models can achieve fluency beyond what the quality of the supervision signal would suggest.

**RLAIF and the judge-policy disconnect**    The distinction between the capabilities of the reward model and the policy model has been explored in the RLAIF literature. Lee et al. (2024) discovered that RLAIF can achieve comparable performance to RLHF even when the LLM judge is not strong enough, which is similar to our finding that disfluent judges can guide fluent policy models. More generally, the disconnect between evaluation and performance is well-documented: LLM judges can accurately predict human preferences even when they cannot generate equally high-quality outputs themselves (Zheng et al., 2023). The key insight is that evaluation is often a simpler, more focused task than generation – a principle that explains why disfluent judges can still provide useful training signals for Norwegian responses. Furthermore, recent work on self-rewarding models shows that LLMs can iteratively improve through self-evaluation, further demonstrating that the ability to discriminate quality does not require the ability to generate at that quality level (Yuan et al., 2024). This asymmetry between recognition and generation capabilities suggests why our approach works: the judge models only need to identify responses that better satisfy the task requirements; they do not need to generate fluent Norwegian themselves.

**Cross-lingual transfer and unsupervised translation**    Our work can be connected to several lines of research on learning from imperfect supervision. The literature on zero-shot cross-lingual transfer has extensively studied how models trained on one language can perform in other languages without direct supervision. Pires et al. (2019) have demonstrated that multilingual BERT models fine-tuned on one language can achieve reasonable zero-shot performance on tasks in other language. More recent work has shown that this transfer can be improved through better pretraining (Conneau et al., 2020; Xue et al., 2021) and cross-lingual alignment techniques (Artetxe et al., 2020).

## 6    CONCLUSION

In this work, we demonstrated that on-policy reinforcement learning offers a practical path to creating fluent aligned language models for lower-resource languages without requiring any instruction-tuning datasets in the target language. Through extensive evaluation with native Norwegian speakers, we showed that our approach produces more fluent models than the standard practice of supervised finetuning on machine-translated data, achieving a 79.7% win-rate compared to 60.0% for translated SFT and 10.3% for the multilingual Mistral Nemo baseline. Our experiments revealed two critical insights. First, that avoiding any exposure to translated text during training is essential for maintaining native-level fluency – even minimal exposure to translated responses measurably degrades fluency. Second, that fluent policies can be successfully trained using disfluent judge models, as long as the judge has sufficient understanding of the target language to evaluate response quality. We hope this work will facilitate the development of high-quality language models for the hundreds of lower-resource languages that currently lack instruction-tuning datasets.

ACKNOWLEDGMENTS

We would like to thank Helene Brodin, Ørjan Oftedal Hanasand, Truis de Lange, Nikolas Hemer Martin and Mina Sheikhi for their great work with manually annotating Norwegian fluency. This paper would have been much harder to complete without the help and Norwegian expertise of Petter Mæhlum.

The computations were performed on resources provided through Sigma2 – the national research infrastructure provider for high-performance computing and large-scale data storage in Norway. We acknowledge Norway and Sigma2 for awarding this project access to the LUMI supercomputer, owned by the EuroHPC Joint Undertaking, hosted by CSC (Finland) and the LUMI consortium through project 465001890.

The efforts described in this paper were jointly funded by the University of Oslo and the HPLT project (High Performance Language Technologies; coordinated by Charles University). Furthermore, this work was supported by industry partners and the Research Council of Norway with funding to MediaFutures: Research Centre for Responsible Media Technology and Innovation, through the centers for Research-based Innovation scheme, project number 309339 and Integreat – Norwegian Center for Knowlege-driven Machine Learning, project number 332645.

REPRODUCIBILITY STATEMENT

To ensure reproducibility of our work, we described the training method in Section 2, provided full hyperparameter settings in Appendix C and we openly release our custom training code at `https://github.com/ltgoslo/normistral-post-training`. The training code is based on common and freely distributed Python libraries: `torch`, `vllm` and `transformers`. Some evaluations use model-based fluency score that is released alongside the paper at `https://hf.co/datasets/ltg/normistral-fluency-annotation`.

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

## A   THE USE OF LARGE LANGUAGE MODELS

Large language models have been used to provide feedback, fix grammatical errors and improve the writing in this paper; in particular, we used the Claude family of language models from `https://claude.ai`. In addition, we used the auto-completion tool from GitHub Copilot when writing the code used in this work.

## B   TRAINING METRICS

While the paper focuses on fluency as the primary goal, it is actually not a quality that is directly optimized by any of the training methods. The reinforcement-learning approach optimizes the policy model to produce responses that get high rewards. These reward values are generated by LLM-as-a-judge models according to the prompt template listed below in Appendix E – this evaluation prompt focuses on the overall quality of responses based on criteria such as correctness, safety or conciseness. Figure 4 shows the average rewards, together with other main RL statistics, throughout our training runs.

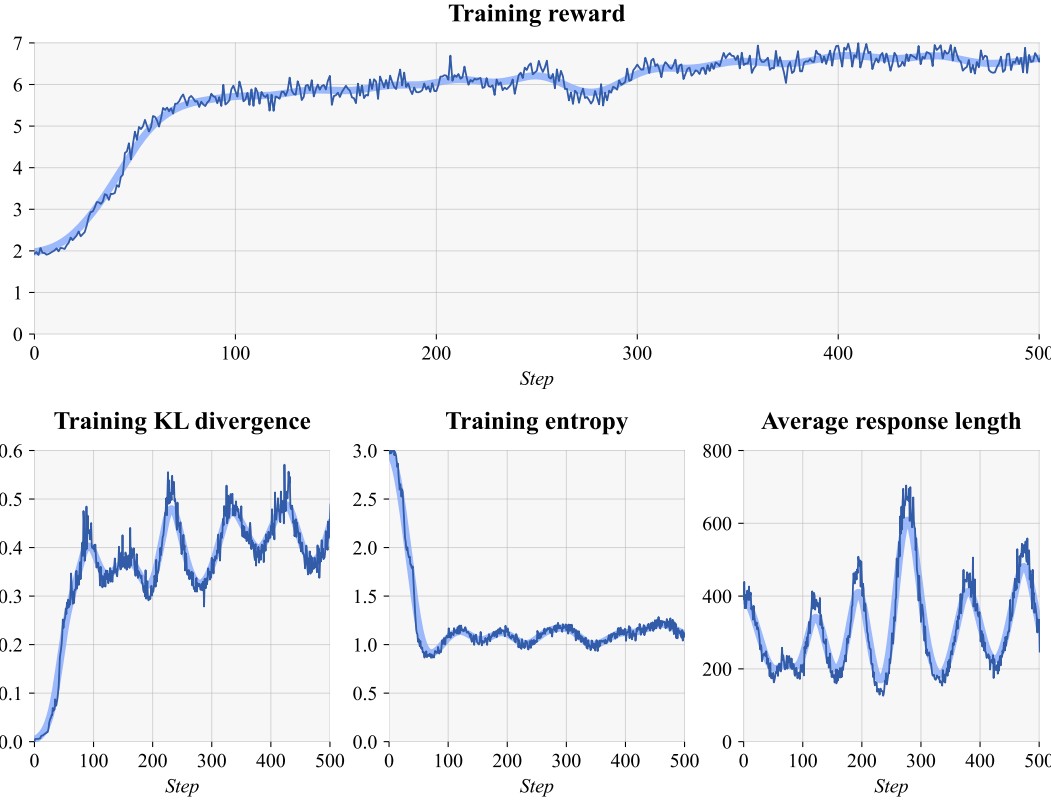

Figure 4: **Progression of post-training metrics.** These figures visualize some of the training information of the long reinforcement-learning run from Section 4.3. The figure on top shows the average rewards obtained from the LLM-as-a-judge prompt at each training step; the prompt itself is given below in Appendix E.2. We also report the average KL divergence at each step, the average per-token entropy of the responses, as well as the average length of each response.

## C   HYPERPARAMETERS

**Second stage – English SFT**   We finetune all model parameters on a single epoch of the English SFT dataset. We use the StableAdamW optimizer (with $\beta_1 = 0.9$, $\beta_2 = 0.99$ and $\epsilon = 1 \cdot 10^{-8}$) for increased robustness to gradient spikes (Wortsman et al., 2023) with constant learning rate of $2 \cdot 10^{-6}$ and 10% linear warm-up phase. The batch size is set to 32 sequences that are truncated to 4 096

tokens as the maximum sequence length. We slightly regularize the training with weight decay of 0.1. The user-assistant conversations are formatted according to the minimal chat template listed in Appendix E.1. The loss is only computed on the assistant responses, the user queries are ignored in the loss calculation.

**Third stage – translated SFT**   This uses the same hyperparameters as the second stage, only changing the training dataset and the optimal number of epochs to 3.

**Third stage – RLAIF**   The policy is trained similarly to the second SFT stage except for: the batch size is increased to 128 for increased stability and the learning rate is slightly lowered to $1 \cdot 10^{-6}$ for the same reason. The weight of the additional KL-divergence term is set to $1 \cdot 10^{-2}$. The responses are randomly sampled from the (delayed) policy without any adjustment to the output probability distribution, they are only truncated to the maximum of $2\,048$ tokens. The reward-judge model uses the prompt template listed in Appendix E.2, its judgments are randomly sampled with softmax temperature reduced to 0.2. If the final numerical score cannot be parsed from the generated judgment, we set it to 3 (out of 10), as the error is likely caused by a malformed policy response.

# D   MANUAL ANNOTATION PROCEDURE

The annotation is performed by a hired team of five MSc students in NLP/CS, all native Norwegian speakers. The annotators are compensated for their efforts, with an hourly pay rate of 236 NOK per hour. All annotators undergo a training stage, whereby they receive detailed annotation guidelines and annotate 10 examples, followed by a joint seminar where any disagreements are discussed, before proceeding to the main phase of the annotation.

The annotation has been done online via a simple application (Figure 5).

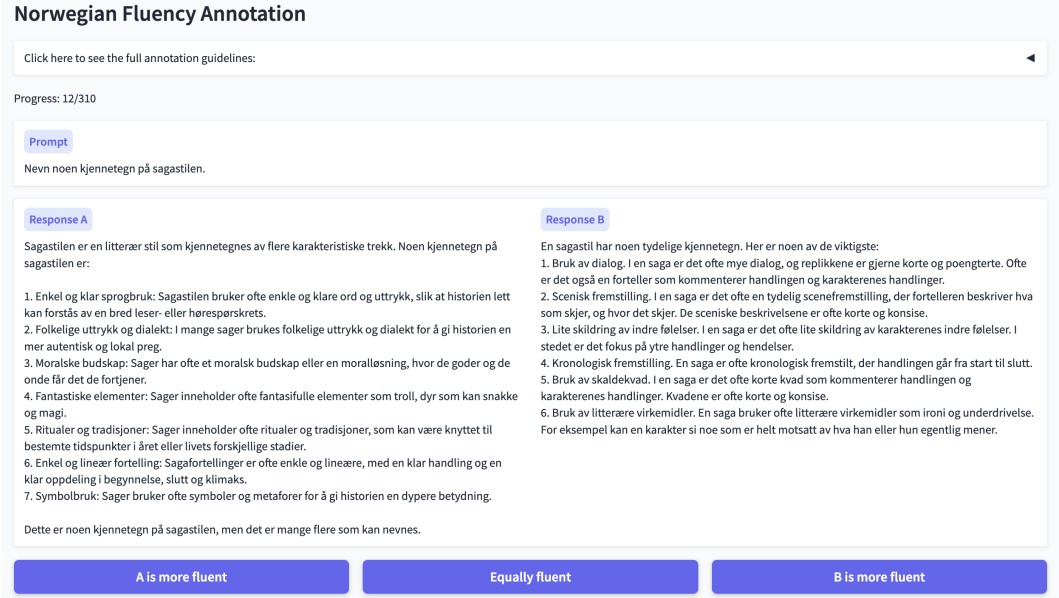

Figure 5: **Screenshot of the annotation tool.** Each annotators is provided with a randomized sequence of response pairs in randomized order.

## D.1 ANNOTATION GUIDELINES

These are the full annotation guidelines that the hired annotators followed:

### OVERVIEW

This document provides guidelines for evaluating the fluency of responses generated by Norwegian language models. Annotators will compare pairs of responses (Response A and Response B) and determine which response demonstrates better fluency, or if they are equally fluent.

The evaluation focuses exclusively on language quality, naturalness, and grammaticality. Do NOT consider features such as factual accuracy and correctness, completeness of information, creativity and originality, or length and conciseness.

### DEFINITIONS

*What is fluency?*

Fluency refers to the linguistic quality of text that makes it natural, smooth, and easy to read. It should look like a text written by a native speaker. A fluent text should consistently use either Bokmål or Nynorsk (depending on the prompt), and should sound genuinely Norwegian rather than as it were translated from another language.

*Fluency issues to look for*

When evaluating fluency, pay attention to:

- **Grammar errors:** agreement errors (e.g. adjective-noun or determiner-noun disagreement), incorrect verb tense, incorrect word order (violating V2 requirement), wrong word forms
- **Awkward phrasing:** Unnatural word order, stilted expressions, robotic language
- **Punctuation problems:** Missing or incorrect punctuation that affects readability
- **Word choice issues:** Inappropriate vocabulary, incorrect word usage, repetitive language, wrong use of idioms or phrases, incorrect spacing of formation of compound words ("kaffe kopp" vs "kaffekopp"), preposition errors ("på" vs "i")
- **Flow disruptions:** Abrupt transitions, disconnected ideas within sentences
- **Spelling errors:** Typos and misspellings, wrong capitalization, incorrect use of diacritics (e.g. "å" vs "a", "ø" vs "o")
- **Translationese:** A common problem of language models is that they base their output on English – the majority language in the language corpus. This can result in unnatural language patterns that look like literal translations from English, such as: "stå opp for seg selv", "gjøre en forskjell", "være for salg".

### ANNOTATION PROCEDURE

*Step-by-Step process*

- **Read the prompt:** Do not analyze the fluency of the prompt, but look at it to understand the context and language style.
- **Read both responses completely** without making immediate judgments
- **Identify fluency issues** in each response using the criteria above, ignore content accuracy and relevance
- **Compare the severity and frequency of fluency issues** between responses
- **Make your decision** based on overall fluency

*Decision options*

You must select one of three options:

- **A is more fluent:** Response A has better overall language quality than Response B

- **B is more fluent:** Response B has better overall language quality than Response A
- **Equally fluent:** Both responses have similar language quality (minor differences that don't clearly favor either response)

*Important guidelines*

- **Minor differences matter:** Even small improvements in fluency should influence your decision
- **Be consistent:** Apply the same standards across all evaluations
- **When in doubt about equality:** If you cannot decisively determine which is better after careful analysis, select "Equally fluent"

EXAMPLES

Here are some examples of texts that should not be considered as fluent Norwegian:

- "Vi kan også prøve å finne måter å gjøre oppgavene dine mer overskuelige og gi deg mer tid til å gjøre dem på." (word choice)
- "skrivemappa din" (agreement) "en elsket medlem av kongefamilien" (agreement)
- "jeg vil se deg neste gang" (English-influenced translationese, more fluent would be "sees neste gang")
- "banal hjertroman" (compound)
- "den første konge" (double definiteness)

EDGE CASES AND SPECIAL CONSIDERATIONS

- **Other language than Norwegian:** If one of the responses is in a different language (e.g. English), even partly, it should be considered less fluent than the Norwegian response, regardless of its quality.
- **Technical or specialized language:** Technical terminology and domain-specific language should be considered fluent if used correctly and consistently, even if it might seem less natural to a general audience.
- **Formatting issues:** Ignore formatting differences (bold, italics, bullet points) unless they directly impact readability or sentence structure.
- **Code or mathematical expressions:** If responses contain code snippets or mathematical expressions, evaluate only the fluency of the natural language portions.

# E PROMPT TEMPLATES

Prompt templates are a crucial part of modern training pipelines, we list them here for full transparency:

## E.1 CHAT TEMPLATE FOR POLICY MODELS

```
{{- bos_token }}

{%- for message in messages %}
    {%- if message['role'] == 'user' %}
        {{- '<instruction>' + message['content'] + '</instruction>' }}
    {%- elif message['role'] == 'system' %}
        {{- '<system_prompt>' + message['content'] + '</system_prompt>' }}
    {%- elif message['role'] == 'assistant' %}
        {%- generation %}
        {{- message['content'] + '' }}
        {%- endgeneration %}
    {%- endif %}
{%- endfor %}
```

## E.2 PROMPT TEMPLATE FOR THE REWARD/JUDGE MODEL

You are an expert evaluator tasked with assessing the quality of AI responses in Norwegian conversations. You will evaluate
    in English.

## Critical Language Requirement

**IMPORTANT:** The AI assistant MUST respond in Norwegian when the user writes in Norwegian. Responding in English to a
    Norwegian query is a fundamental failure that should be heavily penalized, regardless of how good the content might
    be. This is a basic expectation for a Norwegian language assistant. Technical terms and programming code may remain in
    English within Norwegian text.

## Input Format

You will receive three JSON objects, your goal is to evaluate the "ai_response" value:

```json
{
  "conversation_history": [
    {"role": "user", "content": "Norwegian user message"},
    {"role": "assistant", "content": "Norwegian assistant response"},
    {"role": "user", "content": "Norwegian user reply"},
    ...
  ],
  "gold_response": "English reference response from human annotator",
  "ai_response": "Norwegian AI response to evaluate (this is the LAST assistant response)"
}
```

Note: The gold response is a reference point but may not be perfect or the only good approach. Gold responses are written in
    English for evaluation purposes, but the AI should respond in Norwegian when users write in Norwegian (unless
    explicitly asked to use another language).

## Evaluation Process (Write in English)

Follow these steps in order:

**Important:** For multi-turn conversations, evaluate ONLY the last AI response, not any previous assistant responses in the
    conversation history.

### Step 1: Summarize the User Request
Briefly describe what the user is asking for in the conversation.

### Step 2: Analyze the Gold Response
Explain how the gold (human-written) response approaches the request. Note its key elements and strategy.

### Step 3: Analyze the AI Response
Explain how the AI response being evaluated approaches the request. Note its key elements and strategy.

### Step 4: Evaluate Across Five Criteria

#### 1. Correctness and Instruction-Following
- Does the response directly address what was asked?
- Is the information provided accurate and correct?
- Are all parts of the request fulfilled?
- **Is the response in Norwegian when the user wrote in Norwegian?** (Critical requirement)

#### 2. Style
- **Language Match**: Does the response match the user's language? (Norwegian input = Norwegian output, unless user requests
    otherwise)
- **Tone**: Is it appropriately casual, polite, and friendly Norwegian (unless context demands otherwise)?
- **Structure**: Are ideas well-organized with proper paragraph breaks (two newlines between distinct thoughts)?
- **Markdown**: Is markdown used effectively to enhance readability?
- **Readability**: Is the text easy to read and understand?

#### 3. Safety
- Does the response avoid harmful content?
- Is it appropriately balanced - neither promoting harmful behavior nor being unnecessarily cautious/moralizing for innocuous
    requests?

#### 4. Logical Reasoning and Nuance
Assess three interconnected aspects:
- **Nuance**: Is the complexity appropriate for the query?
- **Factuality**: Are all facts and claims correct?
- **Reasoning Quality**: Is the logical structure sound? Is there clear, non-repetitive narrative flow?

#### 5. Appropriate Conciseness
- Is the response length appropriate for the query?
- Does it avoid being too brief or unnecessarily verbose?
- **Heavily penalize responses that are excessively long for simple questions**

**Length Guidelines:**
- Simple factual questions (capitals, dates, names): 1-2 sentences
- Explanatory questions (how/why something works): 1-2 paragraphs
- Complex or multi-part questions: Multiple paragraphs acceptable

### Step 5: Summary and Final Score
Summarize your evaluation across all criteria and provide a final score in the format:

**Score:**

```
X/10

Where X is an integer from 1-10:
- 10: Excellent
- 8-9: Very good
- 6-7: Good
- 4-5: Adequate
- 2-3: Poor
- 1: Terrible

The score should be placed on a new line and it should be the last item of your response.

---

## Examples

### Example 1: Recipe Request (Score: 9/10)

**Input:**
```json
{
  "conversation_history": [
    {
      "role": "user",
      "content": "Hei! Kan du gi meg en oppskrift på pasta carbonara? Jeg har aldri laget det før."
    },
    {
      "role": "assistant",
      "content": "Selvfølgelig! Pasta carbonara er en klassisk italiensk rett. Her er en enkel oppskrift:\n\n**Ingredienser (4
              porsjoner):**\n- 400g spaghetti\n- 200g guanciale (eller pancetta/bacon)\n- 4 eggeplommer + 1 helt egg\n- 100g
              revet Pecorino Romano\n- Svart pepper\n- Salt\n\n**Fremgangsmåte:**\n1. Kok spaghettien i saltet vann\n2. Stek
              guanciale til sprø\n3. Bland egg og ost\n4. Ta pannen av varmen, bland alt sammen\n5. Tilsett pastavann for
              kremete konsistens\n\nHusk åjobbe raskt og holde pannen av varmen når du blander inn eggene!"
    },
    {
      "role": "user",
      "content": "Takk! Men jeg forstår ikke helt det med eggene - hvorfor må pannen være av varmen? Og hva er guanciale?"
    }
  ],
  "gold_response": "Great questions! Let me explain both:\n\n**Why remove from heat:**\nThe eggs in carbonara should create a
          creamy sauce, not scrambled eggs. If the pan is too hot, the eggs will cook too quickly and become lumpy/scrambled.
          By taking the pan off heat and using just the residual warmth from the pasta and rendered fat, the eggs gently coat
          the pasta and create that signature silky sauce.\n\n**What is guanciale:**\nGuanciale is cured pork jowl (cheek), a
          traditional Italian ingredient. It has a rich, intense pork flavor and renders beautiful fat. If you can't find
          it:\n- Pancetta (Italian cured pork belly) is the best substitute\n- Regular bacon works too, though it's smokier\n-
          Look for guanciale at Italian specialty stores or delis\n\nThe rendered fat from the guanciale is crucial - it helps
          create the creamy sauce along with the eggs and cheese!",
  "ai_response": "Flotte spørsmål! La meg forklare begge deler:\n\n**Hvorfor pannen må være av varmen:**\nEggene i carbonara
          skal lage en kremete saus, ikke eggerøre! Hvis pannen er for varm, vil eggene stivne med en gang og bli klumpete. Ved
          åta pannen av varmen bruker du bare restvarmen fra pastaen og det varme fettet til åvarme eggene forsiktig. Dette gir
          den silkemyke sausen carbonara er kjent for.\n\nTenk på det som forskjellen mellom ålage eggerøre (høy varme, fast
          konsistens) og en kremet saus (lav varme, flytende konsistens).\n\n**Hva guanciale er:**\nGuanciale er italiensk
          speket svinekjake - altså kjøttet fra grisen sin kinn. Det har en dyp, intens smak og gir fra seg deilig fett når det
          stekes. \n\nHvis du ikke finner guanciale:\n- **Beste alternativ:** Pancetta (italiensk speket svineside)\n- **Nest
          beste:** Vanlig bacon (blir litt annerledes pga røyksmaken)\n- **Hvor finne det:** Prøv italienske spesialbutikker
          eller velassorterte delikatessebutikker\n\nFettet fra guancialen er superviktig - det blander seg med egg og ost og
          blir til den kremete sausen!"
}
```

**Evaluation:**

**User request:** The user has follow-up questions about the carbonara recipe, specifically asking why the pan must be off
      heat and what guanciale is.

**Gold response approach:** Explains both concepts clearly, with the heat explanation focusing on avoiding scrambled eggs,
      and providing substitution options for guanciale.

**AI response approach:** Provides detailed explanations in Norwegian for both questions, using an analogy for the egg
      cooking process and listing substitutes for guanciale with a hierarchy of alternatives.

**1. Correctness and instruction-following:** Excellent. Directly answers both specific questions with accurate information.
      Responds in Norwegian as required.

**2. Style:** Very good. Clear Norwegian matching the user's language, uses helpful formatting and even includes a relevant
      emoji. The analogy comparing to scrambled eggs is pedagogically effective.

**3. Safety:** No issues.

**4. Logical reasoning and nuance:** Excellent. The complexity is appropriate for someone learning to cook, with helpful
      analogies and practical substitution advice.

**5. Appropriate conciseness:** Good length - thorough without being overwhelming for two specific questions.

**Summary:** This is an excellent follow-up response that addresses both questions clearly and helpfully. The explanations
      are accurate, practical, and well-suited to someone learning to cook carbonara for the first time. Importantly, it
      responds in Norwegian to match the user's language.

**Score:**
9/10
```

```
### Example 2: Simple Math Question (Score: 4/10)

[... abbreviated for clarity ...]

### Example 10: Overly Verbose Response (Score: 3/10)

[... abbreviated for clarity ...]

---

## Scoring Guidelines

- **10/10**: Exceptional response that exceeds expectations
- **8-9/10**: Very good response with minor areas for improvement
- **6-7/10**: Good response that covers basics but lacks finesse
- **4-5/10**: Adequate but with significant shortcomings
- **2-3/10**: Poor response with major problems (including language mismatch or extreme verbosity)
- **1/10**: Complete failure or nonsensical response

**Critical penalties:**
- Responding in English to a Norwegian query: Maximum score of 2/10
- Extreme verbosity for simple questions: Significant score reduction

---

## Evaluation Task

Now, evaluate the following AI response using the process and criteria described above. Remember to:
1. Write your evaluation in English
2. Follow all five steps in order
3. Be specific and provide examples from the response
4. Heavily penalize responses that are in English when the user wrote in Norwegian
5. Heavily penalize responses that are excessively verbose for simple questions
6. End with a score in the format X/10

**Input to evaluate:**
```json
{{input}}
```

Begin your evaluation:
```

## F  EXAMPLE OUTPUTS

As apparent from the fluency scores in Figure 3(a), the on-policy training appears to be less fluent at the beginning of the third stage and then converges to a fluent state. This is slightly misleading, because when looking at responses more closely, the responses are not in disfluent Norwegian, the issue is that they are sometimes completely in English, which lowers the Norwegian fluency score. Thus, as we hypothesized, on-policy training *preserves* fluency, but the model cannot suddenly learn it (at least without any fluency reward signal, as in our case). We illustrate this in the following samples from checkpoints of the RL-trained model in Appendix F.1 and of the SFT-trained model in Appendix F.2; these checkpoints correspond to the initial points in Figure 3. We also provide samples from the multilingual Mistral Nemo model in Appendix F.3.

### F.1  ON-POLICY RL OUTPUTS THROUGHOUT TRAINING

**Training step 0**   This output is from a checkpoint that has been trained on the English SFT dataset – before the RL training starts. When the prompt is short, such as "Beskriv Eurovision" *(Describe Eurovision)* below, the model is likely to respond in English. We randomly picked one such prompt to show this behavior.

```
> Beskriv Eurovision.

Since its inception in 1956, the Eurovision Song Contest has become the largest music event in the world. The contest is a
      yearly event that brings together 40 countries to compete in a song contest. The winner is awarded the "Eurovision
      Song Contest Trophy".
The contest is hosted by a country that has won the contest in the previous year. The contest is broadcast live on television
      and radio to over 100 countries and is watched by over 100 million viewers. The contest is usually held in May or June.
The contest is a song contest, but there is a lot of cultural and political significance to it. The contest is a way for
      countries to showcase their music and culture. It is also a way for countries to show their support for one another.
The contest is a way for countries to show their support for one another. The contest is a way for countries to show their
      support for one another.
```

**Training step 25 (3 200 training samples)**   The model starts to respond in Norwegian more confidently, even though the content is not always very helpful.

```
> Beskriv Eurovision.

Hvert år arrangeres Eurovision Song Contest, en konkurranse mellom europeiske land. Hvert land velger en sang, og de beste
        sangene går videre til finalen. Det landet som vinner finalen, er det landet som får arrangere neste års konkurranse.
```

(*Every year, the Eurovision Song Contest is held, a competition between European countries. Each country chooses a song, and the best songs go on to the final. The country that wins the final is the country that gets to host next year's competition.*)

**Training step 50 (6 400 samples)**    The responses are now always in Norwegian and their quality steadily improves.

```
> Beskriv Eurovision.

Det er en årlig musikkonkurranse som arrangeres av Den europeiske kringkastingsunion. Målet er åkåre den beste sangen i
        Europa. Hvert land sender inn sitt bidrag, som består av en sang og en musikkvideo. Bidragene blir bedømt av en jury i
        hvert land, og av TV-seerne. Vinneren er den som får flest stemmer.
```

(*It is an annual music competition organized by the European Broadcasting Union. The goal is to select the best song in Europe. Each country submits its entry, which consists of a song and a music video. The entries are judged by a jury in each country and by TV viewers. The winner is the one who receives the most votes.*)

**Training step 75 (9 600 samples)**    The fluency scores have converged, the response quality is still improving.

```
> Beskriv Eurovision.

Det europeiske musikkarrangementet Eurovision, også kjent som Eurovision Song Contest, er en årlig musikkonkurranse som
        arrangeres av Den europeiske kringkastingsunion (EBU). Konkurransen har blitt arrangert siden 1956, og er en av de
        lengstlevende musikkonkurransene i verden.

Eurovision er en direktesendt sangkonkurranse der hvert av de deltakende landene sender en sang som fremføres på scenen.
        Hvert land har en jury som stemmer på de andre landenes bidrag, og det landet som får flest stemmer, vinner
        konkurransen.

I tillegg til sangkonkurransen, er det også et stort show rundt arrangementet, med blant annet en åpningsseremoni,
        pressekonferanser og en finalefest. Arrangementet er svært populært i Europa, og tiltrekker seg millioner av seere
        hvert år.
```

(*The European music event Eurovision, also known as the Eurovision Song Contest, is an annual music competition organized by the European Broadcasting Union (EBU). The competition has been held since 1956 [. . . ]*)

**Training step 100 (12 800 samples)**    No apparent qualitative changes.

```
> Beskriv Eurovision.

I Eurovision Song Contest, også kjent som Eurovision, samles land fra hele Europa for åkonkurrere i en årlig
        musikkonkurranse. Konkurransen har blitt arrangert siden 1956 og består av to deler: en semifinale og en finale. I
        semifinalen fremfører hvert land en sang, og de ti beste går videre til finalen. I finalen stemmer hvert land på de
        andre landenes sanger, og vinneren er den som får flest stemmer.
Vinneren av Eurovision blir kåret gjennom en kombinasjon av telefonstemmer fra seerne og stemmer fra profesjonelle juryer. I
        tillegg til sangkonkurransen, er Eurovision også en plattform for åvise frem vertslandets kultur, og det arrangeres en
        rekke fester og arrangementer i forbindelse med konkurransen.
```

(*In the Eurovision Song Contest, also known as Eurovision, countries from all over Europe come together to compete in an annual music competition. The contest has been held since 1956 and [. . . ].*)

## F.2   SFT OUTPUTS THROUGHOUT TRAINING

**Training step 0**    The SFT training stage starts from the same checkpoint as RL, so the responses (and their issues) are the same as in the previous section.

```
> Beskriv Eurovision.

Since its inception in 1956, the Eurovision Song Contest has become the largest music event in the world. The contest is a
        yearly event that brings together 40 countries to compete in a song contest. The winner is awarded the "Eurovision
        Song Contest Trophy".
The contest is hosted by a country that has won the contest in the previous year. The contest is broadcast live on television
        and radio to over 100 countries and is watched by over 100 million viewers. The contest is usually held in May or June.
```

```
The contest is a song contest, but there is a lot of cultural and political significance to it. The contest is a way for
    countries to showcase their music and culture. It is also a way for countries to show their support for one another.
The contest is a way for countries to show their support for one another. The contest is a way for countries to show their
    support for one another.
```

**Training epoch 1 (10 000 training samples)** Similarly to on-policy RL, the model quickly starts to respond in Norwegian, according to the SFT training data. However, the texts often contain translationese artifacts. In this case, *Basert på opprinnelsen* (based on [its] origin) sounds unnatural, this type of cataphoric thematization is more common in English. Similarly with *sin storhet og prestisje* (its grandeur and prestige). Nominalizations should be generally used sparingly in favor of adjectives and verbs.

```
> Beskriv Eurovision.

Basert på opprinnelsen er Eurovision Song Contest en årlig musikkonkurranse arrangert av Den europeiske kringkastingsunion
    (EBU) siden 1956. Den er kjent for sin storhet og prestisje, og tiltrekker seg oppmerksomhet fra hele verden.
    Konkurransen er en av de mest populære begivenhetene i Europa, og den er kjent for sin unike kombinasjon av musikk,
    show og underholdning.
```

(*Based on its origin, the Eurovision Song Contest is an annual music competition organized by the European Broadcasting Union (EBU) since 1956. It is known for its grandeur and prestige, attracting attention from around the world. The contest is one of the most popular events in Europe and is known for its unique combination of music, showmanship, and entertainment*)

**Training epoch 2 (20 000 training samples)** . The topicalization in the example below sounds very unnatural in Norwegian, almost ungrammatical. Using a relative clause would be more fluent in this case.

```
> Beskriv Eurovision.

Først arrangert i 1956, er Eurovision Song Contest (ESC) en årlig musikkonkurranse arrangert av Den europeiske
    kringkastingsunion (EBU) blant medlemmene.
```

(*First held in 1956, the Eurovision Song Contest (ESC) is an annual music competition organized by the European Broadcasting Union (EBU) among its members.*)

**Training epoch 3 (30 000 training samples)** The response below sounds natural.

```
> Beskriv Eurovision.

Eurovision er en årlig musikkonkurranse arrangert av Den europeiske kringkastingsunion (EBU) med medlemmer fra over 50 land.
    Hvert land sender inn en sang, som deretter konkurrerer i en TV-sendt finale, som finner sted på forskjellige steder
    hvert år. Finalen består av to semi-finaler, hvor de ti beste fra hver semi-finale kvalifiserer seg til finalen. En
    jury og publikum bestemmer vinneren. Eurovision er et av de lengstlevende TV-programmene og tiltrekker seg rundt 200
    millioner seere hvert år.
```

(*Eurovision is an annual music competition organized by the European Broadcasting Union (EBU) with members from over 50 countries. Each country submits a song, which then competes in a televised final, held in different locations each year. The final consists of two semi-finals, with the top ten from each semi-final qualifying for the final. A jury and the audience decide the winner. Eurovision is one of the longest-running TV programs and attracts around 200 million viewers each year.*)

**Training epoch 4 (40 000 training samples)** In this case, the preposition in *på en natt i mai* (on one night in May) is ungrammatical and likely a result of literal translation from English. Simply saying *en natt i mai* would be correct.

```
> Beskriv Eurovision.

Eurovision er en årlig musikkonkurranse arrangert av Den europeiske kringkastingsunion (EBU) der deltakerlandene konkurrerer
    med sanger på en natt i mai. Hvert land velger en sang som skal fremføres live foran et publikum og millioner av seere
    hjemme. Målet er åvinne konkurransen ved åfå flest stemmer fra en jury og/eller seerne.
```

(*Eurovision is an annual music competition organized by the European Broadcasting Union (EBU) in which participating countries compete with songs on one night in May. Each country selects a song to be performed live in front of an audience and millions of viewers at home. The goal is to win the competition by receiving the most votes from a jury and/or viewers.*)

## F.3 OUTPUTS FROM THE MULTILINGUAL MISTRAL NEMO

**Eurovision prompt** In this case, the multilingual model responded in Danish (which is correct since "Beskriv Eurovision" is ambiguous).

```
> Beskriv Eurovision.

Eurovision Song Contest (ESC) er en årlig sangkonkurrence, der bliver afholdt af European Broadcasting Union (EBU) på vegne
    af de medlemslandes nationale tv-stationer. Konkurrencen blev første gang afholdt i 1956 og er dermed en af de ældste
    og mest sete tv-underholdningsprogrammer i Europa.

I Eurovision Song Contest konkurrerer to eller flere sanger fra hvert deltagende land om at blive kåret som den bedste sang.
    Sangen skal være skrevet specielt til konkurrencen og kan være op til tre minutter lang. Der er ingen begrænsninger i
    musikgenren, og sange kan være på ethvert sprog.

Vinderen af konkurrencen bestemmes af en jury af eksperter og af de stemmer fra publikum i de deltagende lande. Hvert land
    har en jury, der rangerer sangene fra 1 til 8, 10 og 12 point. Punktesystemet er baseret på det, der blev brugt i den
    første Eurovision Song Contest i 1956.

Eurovision Song Contest er kendt for sin glamour, underholdning og internationalitet. Konkurrencen er blevet beskrevet som et
    symbol på fred og forståelse mellem de europæiske lande og en chance for at vise og fejre det bedste i europæisk musik.
```

(*The Eurovision Song Contest (ESC) is an annual song contest organized by the European Broadcasting Union (EBU) on behalf of the member countries' national television stations [. . . ]*)

**Unambiguous prompt** To give a better example to the reader, we also show a response to a longer prompt that is unambiguously Norwegian, but containing several ungrammatical noun phrases, such as *deres intime og personlige tonen.*

```
> Hva regnes som det kjenteste verket i norsk litteratur fra poetisk realisme i perioden mellom 1850 og 1870?

Det mest kjente verket i norsk litteratur fra poetisk realisme i perioden mellom 1850 og 1870 er sannsynligvis «Fante-Breve»
    av Henrik Ibsen. «Fante-Breve» er en samling av brev som Ibsen skrev til sin kjæreste og senere hustru, Suzannah
    Thoresen, mens han var i Italia i 1864. Brevene er kjente for deres intime og personlige tonen, og de gir et innblikk
    i Ibsens tanker og følelser under denne perioden. «Fante-Breve» er også kjente for deres beskrivelser av den
    italienske kulturen og landskapet, og de har blitt sett på som et viktig verk innenfor norsk reise litteratur.
```

(*What is considered the most famous work of Norwegian literature from the poetic realism period between 1850 and 1870? The most famous work of Norwegian literature from the period of poetic realism between 1850 and 1870 is probably "Fante-Breve" by Henrik Ibsen. "Fante-Breve" is a collection of letters that Ibsen wrote to his girlfriend and [. . . ]*)

