# OpenReview forum: "Fluent Alignment with Disfluent Judges: Post-training for lower-resource languages"
_ICLR.cc/2026/Conference — ICLR 2026 Poster_

### Official Review · Reviewer_9f26 · 2025-10-25

**Soundness:** 3
**Presentation:** 4
**Contribution:** 2
**Rating:** 6
**Confidence:** 2

**Summary:**

The paper aims to assess whether post-training models in low ressource languages for which no natural SFT instruction set exist is better done through (a)  synthetically translating an english SFT set into the target language using a translation model (of varying target language fluency) or (b) using RL to let the target model generate it's own rollouts in the target language and using a **disfluent** reward model to optimize the model towards generating **fluent** text in the target language. In both cases, models are continually pretrained on unsupervised norwegian and lightly SFTed in english beforehand.

Interestingly, this paper shows (b) leads to stronger results, and the target model can become more fluent than the bigger judge model - achieving weak to strong generalization through the combined use of unsupervised CPT of the target model, and the fact that judging fluency for the reward models is an easier task than being fluent in generation themselves.

**Strengths:**

The paper is clearly written. The goals are clearly motivated making reading easy.  The potential impact is clear - enabling simpler training of instruction models on languages where supervised data is scarce.
Ablations are strong and well designed, answering many of the questions I had on the impact of the quality of the translation models for SFT, or judge reward models.

**Weaknesses:**

It would be interesting to have compute intensity comparisons between RL and SFT methods. My intuition is that much more data samples could be seen by the model in the SFT setup under the same compute conditions, perhaps somewhat compensating the performance gap with the RL based method.

One of the big claims of this paper, is that SFT does not work because even with good translation models, the translations may remain unnatural in some low ressource languages. Since the SFT set used is very small, could it have been possible to manually filter the SFT set by norwegian speakers and have a "perfectly fluent" SFT set - thus isolating the RL/SFT debate from translation quality ? More cheaply, a way to compensate the unnaturalness of translated text could also have been to filter translations using the same reward models (maybe generating various rollouts using the translation models and keeping only the most natural as determined by the judges for SFT).  As is, on-policy RL might naturally help the model converge to generations that better appeal to human or model judges irrespective of fluent language.

I would appreciate it if the related work could be slightly extended to include works on translation without/with little parallel data, or works on zero-shot transfer of language abilities (train on an anchor language, evaluate on others). More generally, this problem also maps to works on weak-to-strong generalization, or RLAIF with weak judges and previous papers have already explored this notion of disconnect between judging/doing capacity. I believe discussing this would better contextualize this work and give intuitions on why this is possible.

**Questions:**

1. This setup seems very close to rejection sampling SFT: use the english SFTed model, generate many rollouts in norwegian and train only on the top samples as determined by a LLM as a judge. I'm guessing this might not be possible because post english SFT, the model is not able to produce coherent norwegian so several iterations have to be run? What if this is bootstrapped with few shot ICL in norwegian ? Is RL really key here ?

2. Could the use of techniques such as low-rank adapters, replay data, long learning rate re-warmup, etc. help alleviate the norwegian language collapse when doing english SFT ?

> "The key principle is to never train the language model on any unnatural text."

3. Wouldn't the on-policy sampled text be unnatural at least in the first iterations, even after LLM as a judge filtering ? Could you provide some examples (at least for the norwegian speakers that will understand them, and some qualitative insights into how fluency evolves throughout training, what patterns are learned first, etc) I saw quantitative examples were provided in Figure 3.


My main concern here (summing up most of the weaknesses and question I stated) is disambiguating RL/SFT training dynamics versus the question of training/not training on unnatural samples. Irregardless of if RL is the only way to achieve this, the paper clearly shows judges do not need to be fluent themselves to steer fluent generations which I believe is a nice contribution in this domain.

---

> ### Author Response · Authors · 2025-11-20
>
> We appreciate that the reviewer has carefully gone through our paper and raised valid concerns. We have revised and improved the paper based on the feedback (changes are highlighted in blue, the new sections are in Appendices E and F for now). We also thank the reviewer for acknowledging the impact and robust evaluation of our method.
>
> _____
>
> ***Weakness 1: Compute intensity comparisons between RL and SFT.***
>
> While RL is indeed substantially more compute-intensive than SFT, the key issue is not compute, but the fundamental limitation of training on disfluent data. As Figure 3(a) demonstrates, extended SFT on translated data leads to monotonically decreasing fluency. Even with unlimited compute for SFT, we are bounded by the quality of the training data. In contrast, the RL approach maintains fluency and improves task performance (Figure 3b).
>
> _____
>
> ***Weakness 2: Since the SFT set used is very small, could it have been possible to manually filter the SFT set by norwegian speakers and have a "perfectly fluent" SFT set - thus isolating the RL/SFT debate from translation quality?***
>
> This is a good suggestion; however, there are several fundamental challenges: First, manually creating a "perfectly fluent" SFT set defeats our goal of enabling low-resource language post-training without expensive human annotation. Second, it is already well-established that fluent SFT leads to fluent models. All the existing English models show that and this experiment would not reveal anything new. Third, regarding reward model filtering: our reward models evaluate response quality (correctness, helpfulness, safety), not linguistic fluency. Our method assumes that the judge models are not reliable when it comes to fluency, which is its major benefit.
>
> Overall, the on-policy nature of RL provides a principled solution: the policy model generates responses using its own Norwegian capabilities (developed through pretraining) and only trains on text it has naturally produced.
>
> _____
>
> ***Weakness 3: I would appreciate it if the related work could be slightly extended.***
>
> We completely agree with this point. We have added a new section with extended related work (Appendix E) and we believe that it helps to better put our work into context. We plan to merge this appendix with the Related work section in the main text in the camera-ready version (using the extra page).
>
> _____
>
> ***Question 1: This setup seems very close to rejection sampling SFT.***
>
> You're absolutely right that our approach shares similarities with rejection sampling, but there are crucial differences that make RL more practical and effective here.
>
> Our judge does not evaluate fluency – it assesses correctness, helpfulness, and safety (Appendix D.2). We cannot use it to filter for fluent Norwegian samples because identifying fluency would require precisely the capabilities our judges lack (Table 2). While we used extra annotated data to construct the automatic fluency score, it is used only for evaluation. The training method itself does not assume that the target language has such resources available.
>
> The key, as you correctly note, is the on-policy-ness. RL is a proven method to achieve it in practice.
>
> _____
>
> ***Question 2: Could the use of techniques such as low-rank adapters, replay data, long learning rate re-warmup, etc. help alleviate the Norwegian language collapse when doing English SFT?***
>
> The “Norwegian collapse” is not really an issue, as hopefully better illustrated in the new appendix. The third RL stage quickly learns to consistently output Norwegian. These techniques could indeed help to further mitigate the switch to English, but also complicate the methodology, which we tried to keep straightforward.
>
> _____
>
> ***Question 3: Wouldn't the on-policy sampled text be unnatural at least in the first iterations, even after LLM as a judge filtering? Could you provide some examples?***
>
> As explained in Section 4.2, the problem appears to be switching to English, not unnaturalness. But it is indeed unfortunate to not provide any examples – therefore it's great to receive this feedback! We have now added examples from the initial checkpoints of RL and SFT models, together with examples from the multilingual Mistral Nemo. We also explain the issues in these examples to non-Norwegian speakers. The new section with examples is currently in Appendix F.
>
> _____
>
> ***My main concern here (summing up most of the weaknesses and question I stated) is disambiguating RL/SFT training dynamics versus the question of training/not training on unnatural samples. Irregardless of if RL is the only way to achieve this, the paper clearly shows judges do not need to be fluent themselves to steer fluent generations which I believe is a nice contribution in this domain.***
>
> We hope our previous comments clarified this concern. We are happy to improve the framing in the paper to better address this, if the reviewer believes it would help the reader.

---

> > ### Comment · Reviewer_9f26 · 2025-11-20
> >
> > *On weakness 2*
> >
> > When training your fluency automatic evaluation model, you demonstrate it is possible to create a proxy metric for fluency without relying on fluent judge models (using backtranslation). Beyond this, I imagine (correctness, helpfulness, safety) assessed by the judge are somewhat correlated to fluency (in your prompt you have a readability/style criterion) - maybe the judges are disfluent themselves but still able to assess fluency to a certain degree ?
> > I would still find interesting to see whether relying on such approach to filter the many rollouts of automatic translation methods could improve data quality and alleviate many of the current SFT problems, closing the gap to the RL approach without modifying the setup.
> >
> > Having said that, this is mostly curiosity on my end, and should in no way take away from the solid experiments that are conducted in this paper.
> >
> > **Overall**
> > I believe most of my questions have been clearly answered and I appreciate the modifications made in the manuscript. I believe the paper is rigorous and although not groundbreaking from the RL perspective, I think the paper will be of interest to members of the low ressource language community. I will augment my score but maintain my low confidence score as I am not familiar enough with the current best approaches in RLAIF.

---

### Official Review · Reviewer_dvfW · 2025-11-01

**Soundness:** 3
**Presentation:** 3
**Contribution:** 2
**Rating:** 6
**Confidence:** 3

**Summary:**

This paper proposes an on-policy, fluency-aware post-training pipeline for lower-resource languages. The key idea is: start from a target-language–pretrained model, give it only a short SFT on high-quality English data to learn instruction-following, and then do online, on-policy RL in the target low-resource language — without ever training on machine-translated target-language responses. Even though the reward source (“judge”) can itself be disfluent, the policy can stay fluent because it only ever updates on its own (already fluent) samples.

**Strengths:**

- This paper proposes a three-stage recipe: (1) target-language pretraining (taken as given), (2) short English SFT (1k LIMA items, one epoch) just to teach turn-taking / format, (3) online on-policy RL in the target language, where the policy samples multiple responses per prompt and a multilingual LLM-as-judge gives rewards, plus a KL term with a variance-reduced (Rao–Blackwellized) estimator.
- The experimental section is unusually careful for a “low-resource alignment” paper: same base model lineage, three post-training routes, human A/B judgments (5 annotators, 15–20h each), explicit win-rate table, and multiple ablations (SFT length, translation model, judge choice, training length).
- The pipeline (Fig. 2) and the 3-stage story are easy to follow; section 3 very explicitly says: we tried (1) Norwegian RL, (2) translated SFT, (3) Mistral-Nemo.

**Weaknesses:**

- Scope proven only on one language / one family. Everything is on Norwegian Bokmål, which is relatively resourced compared to truly low-resource or morphologically extreme languages (e.g., Amharic, many Niger-Congo languages). It is also Germanic and fairly close to English. So it’s unclear whether the “disfluent judge → fluent policy” effect would hold for typologically distant languages, or for languages whose MT systems produce much noisier translationese. A small second-language experiment (even synthetic / smaller scale) would strengthen generality.
- The recipe begins with “pretraining on the target language” and even borrows a continuous Norwegian pretraining from Samuel et al. (2025), but the paper explicitly says “we do not cover this pretraining stage.” In practice, the main result (“policy stays fluent”) might be mostly due to starting from a very good Norwegian base — i.e. the method is only as good as your base. The paper could do a clearer disentanglement: what if the base itself is only semi-fluent? The reader can’t currently tell how much credit to give to the RL stage vs the initial base.
- This paper says the judge only needs “sufficient understanding,” not fluency, and Table 2 reports low correlation — but the judge prompt is not analyzed: how sensitive is the whole pipeline to mis-judgments on subtle register/style errors (exactly the kind of thing that makes translationese detectable)? If the judge systematically likes translationese, the policy may reintroduce it despite being on-policy. A robustness test with deliberately biased judges would make the claim stronger.
- Base-model ablation. What happens if the target-language base is weaker (fewer continual-pretraining steps, or a multilingual model zero-shot on Norwegian)? Does the proposed RL still “pull” it to a fluent regime, or is strong pretraining actually the dominating factor?

**Questions:**

- Language Scope and Generality. Could you provide evidence that the proposed “disfluent-judge → fluent-policy” phenomenon holds beyond Norwegian Bokmål? Even a smaller-scale or synthetic experiment on a typologically distant language (e.g., a morphologically rich or non-Indo-European one) would help clarify whether the observed effect generalizes or depends on Norwegian’s proximity to English.
- Dependence on Base-Model Quality. Your pipeline assumes a target-language-pretrained base that is already highly fluent. How much of the final fluency retention actually stems from this strong base versus the on-policy RL stage itself? Could you include an ablation starting from a weaker or multilingual base (zero-shot in Norwegian) to show whether the method can recover fluency or merely preserve it?
- Judge Sensitivity and Bias. This paper states that the judge only needs “sufficient understanding,” yet its potential biases toward translationese or anglicized phrasing are not analyzed. Did you test robustness when the judge systematically prefers slightly translated-style outputs? A stress-test with biased or intentionally noisy judges would clarify how resilient the policy is to imperfect reward signals.
- Please describe the precise prompts and calibration used for the multilingual judge. Are the reward decisions consistent across different prompt wordings or random seeds? Providing examples of disagreements and how they affect updates would make the pipeline more reproducible and interpretable.
- If the base model were only semi-fluent, could the on-policy RL process improve fluency over time, or does it merely avoid degradation? An explicit experiment varying the amount of target-language pretraining would clarify the scope of applicability for teams without high-quality monolingual bases.
- For truly low-resource languages that lack both strong pretraining corpora and fluent judges, what minimal requirements (dataset size, judge quality, reward signal noise tolerance) are necessary for this approach to remain effective? Quantifying those thresholds would make the paper more actionable.

---

> ### Author Response · Authors · 2025-11-20
>
> We would like to thank the reviewer for the constructive feedback.  We are happy for the appreciation of the carefully designed experiments.
>
> First of all, we should clarify that we do not claim that a model can suddenly learn to get fluent during RL (that is most likely not possible, at least without supervision from a fluent judge) – the point is to make sure the policy model does not lose its fluency learned during pretraining. We hypothesize that on-policy post-training can lead to this “fluency preservation” and propose a practical solution that does not require any dataset in the target language.
>
> We view this misunderstanding as a useful feedback for our writing. It has been already highlighted in the first sentence of the abstract (*We propose a post-training method for lower-resource languages that **preserves** fluency of language models even when aligned by disfluent reward models*), but we will do our best to clarify this further in an updated version of the manuscript.
>
> _____
>
> ***Weakness 1: Scope proven only on one language / one family.***
>
> The core mechanism behind our method – that on-policy training preserves fluency even with disfluent judges – is language-agnostic. The principle relies on never exposing the model to unnatural text during training, which should hold regardless of linguistic typology. Our ablation across 8 judge models from different families (Table 2), including models with minimal Norwegian capability, shows consistent fluency preservation (92.1-94.2%), suggesting the robustness of this mechanism. Translationese is a universal phenomenon documented across all language pairs and translation modalities, so the fundamental problem of training on unnatural text remains constant.
>
> Our choice of Norwegian provides a meaningful middle ground: sufficient resources for rigorous evaluation, yet representative of challenges faced by hundreds of languages worldwide. We acknowledge that testing on morphologically complex or non-Indo-European languages would strengthen these claims. However, proper fluency evaluation requires extensive native speaker assessment, making multi-language validation prohibitively expensive while maintaining scientific rigor. We view this as important future work once the principle is established.
>
> _____
>
> ***Weakness 2: Dependence on fluent base model.***
>
> Our method is designed for the realistic scenario where practitioners have access to a fluent base model (from pretraining) but lack instruction-tuning data. This describes the situation for many small languages today. As clarified above, we explicitly acknowledge this dependency throughout the paper. Our method preserves the fluency learned during pretraining rather than creating it. This is a feature, not a limitation – we provide a practical pathway for languages where monolingual pretraining has succeeded but instruction data remains scarce. Our contribution is showing how to maintain fluency through the critical post-training phase. Table 1 demonstrates that even starting from the same fluent base (NorMistral), different post-training approaches yield dramatically different fluency outcomes (79.7% vs 60.0% vs 10.3% win-rates).
>
> We are happy to update the paper to better highlight this important clarification.
>
> _____
>
> ***Weakness 3: Judge prompt and preference of translationese.***
>
> We carefully designed our experimental setup with this concern in mind. The judge prompt (Appendix D.2) explicitly evaluates general response quality across five dimensions without any fluency assessment. This separation is crucial: the judge evaluates helpfulness, safety, and correctness while being blind to fluency considerations.
> If the judges systematically preferred translationese, we would indeed observe fluency degradation in the trained policies. Instead, Table 2 shows remarkably stable fluency (92.1-94.2%) across judges with vastly different capabilities and fluency levels themselves (39.0-84.2%). This stability strongly suggests that fluency preservation emerges from the on-policy mechanism, not from the judge's linguistic preferences.
>
> _____
>
> ***Weakness 4: Base-model ablation.***
>
> We acknowledge that our method requires a fluent base as a prerequisite. This is by design – we address the practical scenario faced by many lower-resource languages that have successful pretraining efforts but lack instruction data. Testing with deliberately weakened bases would not strengthen our claims about fluency preservation; it would test a different hypothesis about fluency generation that we do not make.

---

> > ### Author Response · Authors · 2025-11-20
> >
> > ***Questions 1, 2, 3 and 5.***
> >
> > Addressed by the responses above
> >
> > _____
> >
> > ***Question 4: Please describe the precise prompts and calibration used for the multilingual judge.***
> >
> > We gave the exact prompt template of the judge models in Appendix D.2. The judgments use greedy sampling with temperature 0.0, as common in the RLAIF literature [1]. We did not do any ablation of these hyperparameters or of the prompt, all the 8 tested judge models use the same settings. The self-consistency of the judge rewards has already been studied [1], but it does not seem to greatly impact the performance. Note that RLAIF is an established method and this paper does not aim to prove its general effectiveness.
> >
> > For the benefit of the larger audience that might be reading this discussion, it is worth clarifying that the judge models do not evaluate fluency — the prompt template is by design made to evaluate general response quality regardless of fluency. This is to ensure compatibility of our approach to other RL post-training methods and so that we can use judges that are disfluent. We demonstrated that the fluency remains consistent across judge models with vastly different capabilities, which suggests that it is a general benefit of on-policy training, not of RLAIF.
> >
> > _____
> >
> > ***Question 6: For truly low-resource languages that lack both strong pretraining corpora and fluent judges, what minimal requirements (dataset size, judge quality, reward signal noise tolerance) are necessary for this approach to remain effective?***
> >
> > The minimum requirement is straightforward: a base model with target-language fluency from pretraining. The judge needs only sufficient comprehension to evaluate response quality – Table 2 shows even judges with 39.0% fluency can train 93.1% fluent policies.
> >
> > Quantifying the exact pretraining data requirements is beyond our scope and depends on many factors (language typology, data quality, model size, training compute). We agree that this is an interesting and important question, but our method's value is clear: once you have a fluent base, we provide a path to alignment without the instruction datasets that standard approaches require. This is particularly valuable for languages where monolingual text is available but high-quality instruction data remains scarce – a common scenario for many of the world's languages.
> >
> > _____
> >
> > [1] [RLAIF vs. RLHF: Scaling Reinforcement Learning from Human Feedback with AI Feedback](https://openreview.net/forum?id=AAxIs3D2ZZ) (2024)

---

### Official Review · Reviewer_E2yZ · 2025-11-01

**Soundness:** 1
**Presentation:** 2
**Contribution:** 2
**Rating:** 4
**Confidence:** 3

**Summary:**

This paper investigates on-policy reinforcement learning for improving low-resource language capabilities in multilingual language models. The key insight is both counterintuitive and practical: using a reward model (judge) that is not fully fluent in on-policy RL can outperform supervised fine-tuning (SFT) on machine-translated instruction-following data. Existing multilingual alignment approaches often rely on translating English instruction data, which introduces translationese that harms fluency. In contrast, on-policy reinforcement learning can achieve alignment purely through model-generated rollouts, without relying on synthetic or translated data. However, this requires a robust reward model to guide the RL process. The authors show that such a reward model need not be fluent in the target language. Using Norwegian as a case study, the authors conduct human evaluations by five native speakers, demonstrating that on-policy RL guided by a disfluent judge achieves higher fluency compared to translation-based SFT baselines.

**Strengths:**

1. The exploration of RL for low-resource language alignment is novel and practically meaningful. It avoids annotation costs and translation artifacts, and shows that even a non-fluent reward model can guide the training fluent policy models in the target language.
2. The use of human evaluation by native speakers strengthens the credibility of the findings. The results in Table 1 clearly demonstrate the superiority of the proposed approach.
3. Beyond using a non-fluent judge, the paper further explores different model families as reward models and shows that the final alignment quality in RL is independent of the judge model’s fluency.

**Weaknesses:**

1. Generalization to other languages. All experiments are conducted solely on Norwegian. It remains unclear whether the proposed method generalizes to other low-resource languages or even to medium- and high-resource ones. Would the same conclusions hold across different language families? For higher-resource languages, might the performance gap with SFT narrow? While human evaluation is costly, automated evaluation (similar to the constructed evaluation model in the paper) could help extend this analysis.
2. Evaluation scope limited to fluency. Fluency is central to the paper’s evaluation, but other aspects such as helpfulness [1] or safety are equally important. There is a risk of reward hacking [2]: the model might prioritize fluency at the cost of informativeness, producing simpler but less useful responses. It remains unclear whether fluency gains come at the expense of other abilities.
3. Dependence on SFT for initialization. The Norwegian RL pipeline still relies on some SFT data to establish basic instruction-following ability. The claim would be more solid if the authors demonstrated pure on-policy RL starting from a base model, similar to the success of zero RLVR [3], showing that RL alone can enhance the model’s capabilities without prior supervised alignment.
4. Limited model diversity. The experiments are conducted mainly with the Mistral base model. Extending the study to additional model families (e.g., Qwen, Llama) would strengthen the generality and robustness of the conclusions.
5. The section on human evaluation lacks important background information about the annotators. To ensure ethical research practices, the authors should specify the annotators' qualifications and confirm that they received fair and reasonable compensation for their work.

---

**References**

[1] Liu, Ryan, Theodore Sumers, Ishita Dasgupta, and Thomas L. Griffiths. (2024) "How do Large Language Models Navigate Conflicts between Honesty and Helpfulness?." In International Conference on Machine Learning, pp. 31844-31865. PMLR.

[2] Liu, Tianqi, et al. (2024) "RRM: Robust Reward Model Training Mitigates Reward Hacking." The Thirteenth International Conference on Learning Representations.

[3] Guo, D., Yang, D., Zhang, H., Song, J., Zhang, R., Xu, R., ... & He, Y. (2025). Deepseek-r1: Incentivizing reasoning capability in llms via reinforcement learning. arXiv preprint arXiv:2501.12948.

**Questions:**

1. Where does the boundary of the main claim lie? i.e., What is the minimum level of language proficiency required for a reward model to provide effective reward signals? For instance, could a strong English model with only weak, translation-based understanding of Norwegian still serve as an effective judge?

---

> ### Author Response · Authors · 2025-11-20
>
> We thank the reviewer for the thorough and constructive review. We appreciate the recognition of our work's novelty and practical significance, particularly highlighting the counterintuitive finding that disfluent judges can guide fluent policy training. Below, we address each concern raised and provide additional clarification where needed.
>
> _____
>
> ***Weakness 1: Generalization to other languages.***
>
> We chose Norwegian precisely because it represents a challenging case: a lower-resource language without any instruction-tuning data, which makes it ideal for demonstrating the effectiveness of our method.
>
> Translationese is a universal phenomenon, not specific to Norwegian. It is present even in professional human translations and machine translation amplifies these artifacts regardless of resource level [1, 2]. As shown in Table 4, while better translation models reduce the fluency gap, they cannot eliminate it – the fundamental issue of training on unnatural text remains. There is no evidence that translationese stops being an issue for higher-resource languages.
>
> We completely agree that extending to more languages would be valuable future work. However, when we designed the experiments, we paid a lot of attention to properly evaluate the fluency by native speakers, as purely automatic fluency evaluation is a difficult problem; extending the experiments to more languages while maintaining the quality would be prohibitively expensive.
>
> We believe our thorough evaluation on Norwegian, combined with the universal nature of translationese, provides strong evidence for the broader applicability of our method.
>
> _____
>
> ***Weakness 2: Evaluation scope limited to fluency.***
>
> We understand that there is a possibility of some performance trade-off and it is a valid concern. However, our RL method does not explicitly optimize for fluency: the judge prompt (Appendix D.2) evaluates across five dimensions: (1) correctness & instruction-following, (2) style, (3) safety, (4) logical reasoning & nuance, and (5) appropriate conciseness. We specifically designed it this way so that it reflects a practical RLAIF post-training setup. Fluency is preserved indirectly through on-policy training, not through reward optimization.
>
> This design choice directly addresses the reviewer's concern: since our reward signal already includes safety and helpfulness, there's no risk of the model "prioritizing fluency at the cost of informativeness." The fluency gains emerge from never training on disfluent text, while the reward signal ensures the model maintains or improves on other important dimensions. Our evaluations support this: Figure 3b shows NLG performance improving throughout RL training alongside fluency.
>
> We agree with the concern that responses might appear more fluent just because they are simpler, we will clarify this in the camera-ready version. The RL-trained model isn't producing "simpler but less useful responses". The updated manuscript now contains generated responses throughout RL training that show this (Appendix F). The average response lengths in the main experiment (Table 1) are similar – on-policy RL: 196.8 words, translated SFT: 198.1 words, and Mistral Nemo: 240.3 words.
>
> _____
>
> ***Weakness 3: Dependence on SFT for initialization.***
>
> Our choice of minimal English SFT (just 1000 openly available LIMA examples) is deliberate and practical. As shown in Table 3, this minimal initialization is sufficient and, crucially, doesn't introduce disfluencies since we never train on translated responses.
> While pure RL from base models is intellectually interesting, the standard approach in modern LLMs is to “cold-start” from an SFT dataset, which is what we followed [3].  We believe this makes our work more relevant in practice. Note that DeepSeek R1, which the reviewer mentions, also uses SFT initialization in their final pipeline.
>
> The key insight isn't about eliminating SFT entirely, but showing that fluency can be maintained by never training on translated text during the critical alignment phase.
>
> _____
>
> ***Weakness 4: Limited model diversity.***
>
> The core point – that on-policy RL maintains fluency better than training on translated data – is based on fundamental principles about exposure to unnatural text, not architecture-specific details.
>
> We chose NorMistral as it's the only publicly available Norwegian-pretrained model of sufficient scale and capability [4]. Creating comparable Norwegian versions of Qwen or Llama would require months of pretraining and substantial computational resources, well beyond the scope of a rebuttal period. However, our ablations across different judge models (Table 2, including Qwen and Llama variants) provide evidence that the approach works across model families of judge models.

---

> > ### Author Response · Authors · 2025-11-20
> >
> > ***Weakness 5: The section on human evaluation lacks important background information about the annotators.***
> >
> > Absolutely, we agree with the reviewer about the importance of research ethics. This information was already provided in Appendix C (referenced on line 289): "The annotation is performed by a hired team of five MSc students in NLP/CS, all native Norwegian speakers. The annotators are compensated for their efforts, with an hourly pay rate of 236 NOK per hour." We also detail their training process, including practice annotations and calibration seminar in that appendix.
> >
> > ***Question 1: Where does the boundary of the main claim lie? For instance, could a strong English model with only weak, translation-based understanding of Norwegian still serve as an effective judge?***
> >
> > Excellent question. Table 2 directly addresses this: even Qwen 2.5 72B, with Norwegian fluency score of 50.7% (substantially worse than our translated baseline at 85.7%), successfully trains a policy with 92.9% fluency. This suggests the boundary is quite permissive – the judge needs sufficient comprehension to evaluate response quality but not native-level fluency. The key is that the judge understands the task semantics, even if its own generation would be disfluent.
> >
> > _____
> >
> > [1] [How Human is Machine Translationese? Comparing Human and Machine Translations of Text and Speech](https://aclanthology.org/2020.iwslt-1.34/) (2020)
> >
> > [2] [Lost in Translation: Loss and Decay of Linguistic Richness in Machine Translation](https://aclanthology.org/W19-6622/) (2019)
> >
> > [3] [The Smol Training Playbook: Beyond base models — post-training in 2025](https://huggingfacetb-smol-training-playbook.hf.space/#beyond-base-models--post-training-in-2025) (2025)
> >
> > [4] [Small Languages, Big Models: A Study of Continual Training on Languages of Norway](https://aclanthology.org/2025.nodalida-1.61/) (2025)

---

### Meta-Review · Area_Chair_6Unr · 2026-01-04

**Summary:**

The paper proposes an approach to improve quality LLMs in low resource settings, using a weak LLM as judge for on-policy RLAIF. The reviewers agree that the results are surprising and practical, given an LLM that’s not fluent in a language can be used to improve quality over using machine-translated data for RL.

The main issues raised during the review:
1. Experiments are limited to Norwegian. Unclear how well the technique generalizes to other low resource languages, or medium- high- resource languages and tasks. Relative similarities to other high resource languages also make the results somewhat limiting, when it comes to generalization to more unique low-resource languages.
2. Evaluations are limited (only fluency, not other aspects of quality).
3. Limited to one model (Mistral).
4. Unclear how bad the judge LLM can be to get good performance. Similarly, unclear how good the pre-trained model has to be to get good quality in the end.
5. Additional analysis on how the prompt / fluency affects quality.
6. Using the weak judges to filter SFT translated data, as another baseline.

**Reviewer Concerns:**

Overall, the main issue is that the evals are limited to one language and one model architecture. Although the authors argue that the technique itself is agnostic, without additional experiments, it’s hard to know whether it’ll generalize. Therefore, the paper is borderline at the moment.

1. The authors argue that evaluations, since it involves humans and is not automatic, is prohibitive and expensive. They further argue that translationese is generally hard irrespective of the amount of data, so their hypothesis is that the technique should transfer. The answer here is a bit handwavy and unsatisfactory. For high resource languages, perhaps use of automated evals is still fine. As of now, it’s hard to assess how general the results are.
2. The authors argue that their judge scores across multiple dimensions, not just fluency. But again, results across other dimensions are missing.
3. The authors argue that their main hypothesis is independent of the model architecture. While true, this is again handwavy, and additional results would have strengthened the case. They also argue that Mistral is the only model pre-trained with Norwegian. Having results on additional languages and models would have helped all the more in this case.
4. The authors show results with Qwen as the judge, which has a bad fluency score. So the method seems to generalize reasonably well. Would have been useful to quantify this in some way. As for the fluency of the pretrained model, the authors clarify that their method aims to “preserve” fluency of the pretrained model. They also say that data requirements are beyond the scope of the work (correctly so), but perhaps the reviewer was asking a different question (how good should the pretrained model be, or how to quantify it?).
5. The authors commented that they did not do any ablations with the prompt.
6. The authors argue that creating fluent data is hard; but again, it does not seem like the point the reviewer was getting, as they clarify later. The authors did not provide additional clarification to this.

**Reviewer Scores:**

Reviewer E2yZ (Runzhe Zhan): 4 -> 4

Reviewer dvfW (Xiaolong Wang): 6 -> 4/6

Reviewer 9f26 (Manuel Faysse): 6 -> 6

---

### Decision · Program_Chairs · 2026-01-26

Accept (Poster)